# SLA: Beyond Sparsity in Diffusion Transformers via Fine-Tunable Sparse–Linear Attention

**Jintao Zhang**[12], **Haoxu Wang**[1], **Kai Jiang**[1], **Shuo Yang**[2], **Kaiwen Zheng**[1], **Haocheng Xi**[2], **Ziteng Wang**[1] **Hongzhou Zhu**[1], **Min Zhao**[1], **Ion Stoica**[2], **Joseph E. Gonzalez**[2], **Jianfei Chen**[1], **Jun Zhu**[1*]
[1]Dept. of Comp. Sci. & Tech., BNRist Center, Tsinghua-Bosch ML Center, Tsinghua University
[2]UC Berkeley
{zhang-jt24, jianfeic, dcszj}@tsinghua.edu.cn

## Abstract

In Diffusion Transformer (DiT) models, particularly for video generation, attention latency is a major bottleneck due to the long sequence length and the quadratic complexity. Interestingly, we find that attention weights can be decoupled into two matrices: a small fraction of large weights with high rank and the remaining weights with very low rank. This naturally suggests applying sparse acceleration to the first part and low-rank acceleration to the second. Based on this finding, we propose SLA (**S**parse-**L**inear **A**ttention), a trainable attention method that fuses sparse and linear attention to accelerate diffusion models. SLA classifies attention weights into critical, marginal, and negligible, applying $\mathcal{O}(N^2)$ attention to critical weights, $\mathcal{O}(N)$ attention to marginal weights, and skipping negligible ones. SLA combines these computations into a single GPU kernel and supports both forward and backward passes. With only a few fine-tuning steps using SLA, DiT models achieve about $20\times$ reduction in attention computation, resulting in significant acceleration without loss of generation quality. Experiments show that SLA reduces attention computation by about $95\%$ without degrading end-to-end generation quality, outperforming baseline methods. In addition, we implement an efficient GPU kernel for SLA, which yields a $13.7\times$ speedup in attention computation and a $2.2\times$ end-to-end speedup in video generation on Wan2.1-1.3B. The code is available at https://github.com/thu-ml/SLA.

## 1 Introduction

Among the operations in Transformers, attention (Vaswani et al., 2017) is the only one with quadratic computation complexity, while others mostly scale linearly with the sequence length $N$. In Diffusion Transformer (DiT) models (Peebles & Xie, 2022), especially for video generation, attention becomes the primary computational bottleneck, as the sequence length typically ranges from 10K to 100K. Reducing the cost of attention is therefore critical for improving the efficiency of DiT models. Existing efficient attention methods for DiTs fall into two main categories: (1) numerous *sparse attention* methods (Li et al., 2025; Zhang et al., 2025b; Xi et al., 2025; Yang et al., 2025a; Zhang et al., 2025e; Wu et al., 2025; Shen et al., 2025; Hassani et al., 2023; Liu et al., 2025), which compute only a subset of attention scores, and (2) a few *linear attention* methods (Xie et al., 2024; Zhu et al., 2025), which reformulate the operation to achieve $\mathcal{O}(N)$ complexity.

**Limitation.** Despite recent progress, both approaches face challenges in substantially reducing attention computation: **(L1)** Linear attention methods often fail in practice, especially on video diffusion models. Existing work on linear attention in diffusion is rare and primarily limited to image generation. Our experiments show that when applied to diffusion models, particularly video generation, linear attention severely degrades video quality. **(L2)** Sparse attention methods rarely achieve very high sparsity and require a considerable fraction of the full complexity of attention. In practice, they typically reach only 40–60% sparsity for sequence length below 50K. Although some recent works (Yang et al., 2025a; Li et al., 2025) report sparsity of 80–85%, such results are obtained on very long sequences (e.g., 100K–300K), where achieving high sparsity is easier.

---

*Corresponding author.

**Key Observation.** We find that attention weights in diffusion transformers can be decomposed into two matrices: a small fraction of large weights with high rank and a large fraction of the remaining weights with extremely low rank. This explains why sparse attention or linear attention alone cannot achieve satisfactory results and naturally suggests applying sparse acceleration to the first part and low-rank acceleration to the second.

**Our Method.** Based on the observation above, we propose SLA, a trainable hybrid sparse and linear attention for DiT models. Specifically, attention weights are partitioned into blocks and dynamically classified into three categories: critical, marginal, and negligible. Critical blocks are computed exactly using FlashAttention, negligible blocks are skipped, and, unlike existing methods, marginal blocks are processed with linear attention. This design allows sparsity to increase dramatically (e.g., 70%→95%) while maintaining accuracy. Since linear attention is computationally negligible, costing less than 0.5% of full attention in video generation models, SLA is several times faster than sparse attention alone. Furthermore, we implement efficient forward and backward passes for SLA. With a few steps of fine-tuning, SLA significantly reduces the computation complexity and latency of attention while preserving the quality of the generation results.

**Result.** SLA reduces attention computation by about **95%** without degrading video generation quality, even at a moderate sequence length of 30K, which is the sequence length in Wan2.1-1.3B. In addition, our implementation achieves a **13.7×** speedup in the attention kernel and a **2.2×** end-to-end acceleration for video generation, where the attention time becomes almost negligible. SLA consistently surpasses baselines in both generation quality and efficiency.

**Contribution.** We summarize our main contributions as follows. (1) We find that the attention weights in diffusion models can be perfectly decomposed into two parts: a highly sparse matrix with high rank and a dense matrix with very low rank. (2) We propose the first attention method that effectively fuses sparse attention and linear attention. (3) Our method achieves about 95% attention sparsity, corresponding to approximately a 20× reduction in attention computation, while maintaining video generation quality. (4) We implement efficient GPU kernels for SLA.

## 2 PRELIMINARY

### 2.1 BLOCK SPARSE ATTENTION

Given queries, keys, and values $Q, K, V \in \mathbb{R}^{N \times d}$, the standard attention computes the score matrix $S = QK^\top / \sqrt{d}$ and the attention weights $P = \text{Softmax}(S)$ to obtain the output $O = PV$. This is inefficient for large $N$ as it requires $\mathcal{O}(N^2 d)$ operations. The idea of sparse attention is to reduce computation by applying a mask $M \in \{0, 1\}^{N \times N}$ to the attention weights: $P \leftarrow P \odot M$, where $\odot$ is the element-wise product. A common strategy is to choose a threshold $\tau$ and set $M_{ij} = 1$ if $P_{ij} > \tau$. For entries with $M_{ij} = 0$, the multiplications $Q_i K_j^\top$ and $P_{ij} V_j$ can be skipped, where $Q_i = Q[i, :], K_j = K[j, :], V_j = V[j, :]$.

However, element-wise sparse attention is inefficient on modern GPUs. Practical implementations such as FlashAttention (Dao, 2023) operate at the block level. Specifically, the sparse FlashAttention first partitions $Q, K, V, S, P, M$ into blocks $\{\mathbf{Q}_i\}, \{\mathbf{K}_j\}, \{\mathbf{V}_j\}, \{\mathbf{S}_{ij}\}, \{\mathbf{P}_{ij}\}, \{\mathbf{M}_{ij}\}$, where $\mathbf{Q}_i \in \mathbb{R}^{b_q \times d}$, $\mathbf{K}_j, \mathbf{V}_j \in \mathbb{R}^{b_{kv} \times d}$, and $\mathbf{S}_{ij}, \mathbf{P}_{ij}, \mathbf{M}_{ij} \in \mathbb{R}^{b_q \times b_{kv}}$. Each block mask $\mathbf{M}_{ij}$ is fully filled with either 0 or 1, and we skip the computations of $\mathbf{Q}_i \mathbf{K}_j^\top$ and $\mathbf{P}_{ij} \mathbf{V}_j$ if $\mathbf{M}_{ij}[:, :] = 0$.

### 2.2 LINEAR ATTENTION

Linear attention methods reduce the complexity of standard attention from $\mathcal{O}(N^2 d)$ to $\mathcal{O}(N d^2)$. A key idea is to decouple the softmax operation by introducing a feature map $\phi(\cdot)$ applied to $Q$ and $K$. Specifically, it replaces the attention weights in standard attention with $\frac{\phi(Q)\phi(K)^\top}{\text{rowsum}(\phi(Q)\phi(K)^\top)}$. This reformulation enables reordering of the matrix multiplications: instead of explicitly computing the attention weights, it first computes $\phi(K)^\top V$, and then applies this intermediate result to $\phi(Q)$:

$$H = \phi(K)^\top V, \quad Z = \text{rowsum}(\phi(K)^\top) \in \mathbb{R}^{d \times 1}, \quad O = \frac{\phi(Q)H}{\phi(Q)Z}.$$

The mapping $\phi(\cdot)$ is usually an activation function (e.g., ELU + 1 or ReLU (Clevert et al., 2016; Xavier et al., 2011)). This formulation avoids explicitly constructing the $N \times N$ matrices $S, P$ and achieves linear computational complexity.

# 3 MOTIVATION AND ANLYSIS

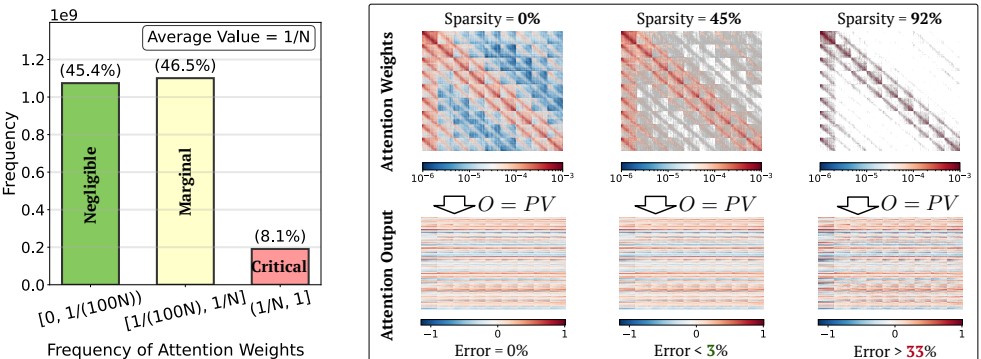

Figure 1: The left figure shows a typical distribution of attention weights sampled from the Wan2.1 model. The right figure shows the accuracy of sparse attention with different sparsity.

## 3.1 MOTIVATION OF SLA

Due to the softmax operator, the attention weights $P$ lie in $[0, 1]$ with each row summing to 1. Furthermore, because of the exponential scaling in softmax, only a small fraction of entries in $P$ are relatively large, while the vast majority are close to zero. Figure 1 (left) shows the typical distribution of attention weights $P$ sampled from the Wan2.1 model (Wan et al., 2025). We highlight two key observations: (1) Only about $8.1\%$ of the weights are larger than the average value $1/N$. (2) A considerable proportion of weights are extremely small. In our case, approximately $45\%$ fall below $1/(100N)$. As shown in Figure 1 (right), skipping these smallest $45\%$ of weights in sparse attention (i.e., setting the corresponding entries in $M$ to 0) introduces a relative L1 error of less than $3\%$ compared to the full attention output. In contrast, retaining only the largest $8.1\%$ of weights (sparsity $= 92\%$) leads to a sharp increase in error, reaching about 33%. This explains why existing sparse attention methods struggle to achieve a sparsity beyond 90%.

The intermediate values between $1/(100N)$ and $1/N$ (the yellow column in Figure 1) present a dilemma: omitting them introduces significant accuracy loss, yet computing them with full attention causes a great decrease in sparsity. Fortunately, these values are far less critical than the largest ones. This finding motivates us to categorize the attention weights into three types: *critical*, *marginal*, and *negligible*. For *critical* weights, we use sparse FlashAttention to compute the output as they dominate the attention distribution; For *negligible* weights, we skip the computation; For *marginal* weights, we employ a linear attention method to reduce the computational complexity to $\mathcal{O}(Nd^2)$ and enhance the performance of sparse attention.

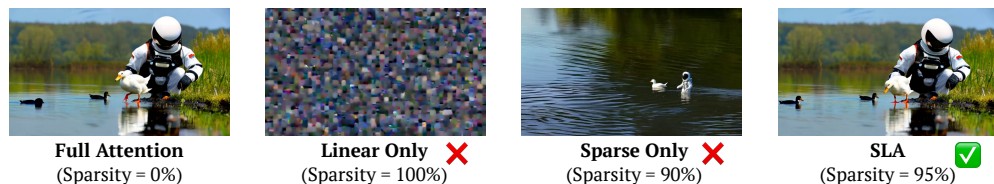

Figure 2: Video generation examples on Wan2.1 fine-tuned with full attention, linear attention, sparse attention, and SLA. SLA could achieve a high sparsity of 95% and lossless video quality.

Empirical results. In Figure 2, we present some videos generated by Wan2.1 fine-tuned with different attention methods: using only linear attention, sparse attention with 90% sparsity, and SLA with 95% sparsity. Note that the computational complexity of SLA at 95% sparsity is nearly half that of 90% sparse attention, since the cost of linear attention is almost negligible. For example, in the Wan2.1 model, linear attention accounts for less than 0.5% of the cost of full attention. These empirical results show that SLA significantly outperforms the other two methods in video quality.

## 3.2 SEPARATING ATTENTION WEIGHTS: SPARSE FEW, LOW-RANK MANY

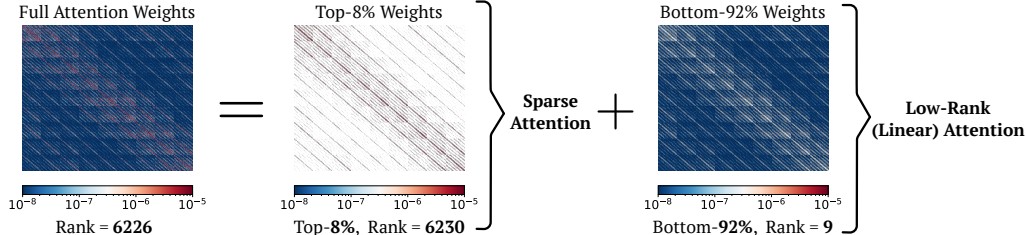

Figure 3: Decomposition of attention weights. We sample attention weights from the Wan2.1 model: the left figure shows the full weights, the middle the top 8%, and the right the bottom 92%.

**Observation.** *As shown in Figure 3, full attention weights can be decoupled into two parts: (1) a small subset ($< 10\%$) with rank comparable to full attention, and (2) a large subset ($> 90\%$) with very low rank. Since the methods for accelerating attention focus mainly on sparsity or low-rank structure, this suggests a **natural and elegant strategy**: apply sparse attention to the first part and low-rank approximation to the second.*

Previous failures of linear attention are largely due to the high rank of full attention weights (Fan et al., 2025), while linear attention is restricted to a rank at most $d$. Figure 3 (left) illustrates this with a typical example using the notion of stable rank (Rudelson & Vershynin, 2006). We observe that after removing the top values in the attention weights $P$, the remaining matrix becomes extremely low-rank. This motivates the decomposition of $P$ using the sparse mask $M$:

$$P = \underbrace{P \odot M}_{\text{sparse component}} + \underbrace{P \odot (1 - M)}_{\text{low-rank component}}. \tag{1}$$

Since linear attention is essentially a low-rank version of attention, we are provided with a possibility to replace the low-rank component $P \odot (1 - M)$ with linear attention.

## 4 SLA

SLA effectively integrates sparse and linear attention within a unified framework, allowing them to complement each other. In particular, we fuse both attention into a single efficient GPU kernel. In this section, we introduce the sparse and linear attention components of SLA.

SLA first predicts a compressed attention weights matrix $P_c \in \mathbb{R}^{N/b_q \times N/b_{kv}}$:

$$P_c = \text{Softmax}(\text{pool}(Q)\text{pool}(K)^\top/\sqrt{d}). \tag{2}$$

where $\text{pool}(\cdot)$ is a mean pooling operator along the token dimension. For each element of $P_c$, we classify it into three types and record the results in a compressed mask $M_c \in \mathbb{R}^{N/b_q \times N/b_{kv}}$. Specifically, the top $k_h\%$ positions are marked as critical (labeled 1), the bottom $k_l\%$ positions as negligible (labeled $-1$), and the remaining positions as marginal (labeled 0). Formally,

$$M_c[i,j] = \{1 \text{ (top } k_h\%), \quad -1 \text{ (bottom } k_l\%), \quad 0 \text{ (otherwise)}\}. \tag{3}$$

We apply different methods according to $M_c$.

### 4.1 SPARSE ATTENTION IN SLA

Guided by the mask $M_c$, sparse FlashAttention is used to compute the sparse attention output. For each $Q$ block $\mathbf{Q}_i$, we iterate over all $K, V$ blocks $\mathbf{K}_j, \mathbf{V}_j$ with $j = 0, \dots, N/b_{kv}$. Whenever $M_c[i,j] = 1$, we perform:

$$\mathbf{S}_{ij} = \mathbf{Q}_i \mathbf{K}_j^\top/\sqrt{d}, \quad \mathbf{P}_{ij} = \text{OnlineSoftmax}(\mathbf{S}_{ij}), \quad \mathbf{O}_i^s = \mathbf{O}_i^s + \mathbf{P}_{ij}\mathbf{V}_j. \tag{4}$$

Here, $\text{OnlineSoftmax}(\cdot)$ operator (Milakov & Gimelshein, 2018) computes the softmax of a matrix in a block-wise manner (see lines 10-11 of Algorithm 1 for implementation). The initial value of each $\mathbf{O}_i^s$ is set to zero. Algorithm 1 describes the forward computation of the sparse attention component, and we denote the final output of the sparse attention component $O^s$.

## 4.2 LINEAR ATTENTION IN SLA

Inspired by the idea of low-rank approximation, we replace the low-rank component $P \odot (1 - M)$ in Equation 1 with linear attention introduced in Section 2.2 as

$$\frac{\phi(Q)\phi(K)^\top}{\mathrm{rowsum}(\phi(Q)\phi(K)^\top)} \odot (1 - M).$$

Specifically, the entries of 0 in $M_c$ determine the blocks processed by linear attention. For each query block $\mathbf{Q}_i$, we compute the corresponding linear attention output:

$$\mathbf{H}_i = \sum_{j:M_c[i,j]=0} \phi(\mathbf{K}_j)^\top \mathbf{V}_j, \quad \mathbf{Z}_i = \sum_{j:M_c[i,j]=0} \mathrm{rowsum}(\phi(\mathbf{K}_j)^\top), \quad \mathbf{O}_i^l = \frac{\phi(\mathbf{Q}_i)\mathbf{H}_i}{\phi(\mathbf{Q}_i)\mathbf{Z}_i}. \quad (5)$$

Here, as mentioned in Section 2.2, $\phi(\cdot)$ denotes the activation function, and $\mathbf{H}_i \in \mathbb{R}^{d \times d}, \mathbf{Z}_i \in \mathbb{R}^{d \times 1}$ are intermediate results similar to $H$ and $Z$. Algorithm 1 describes the forward pass of the linear attention component, and the final output of this component is denoted as $O^l$.

Finally, the overall attention output of SLA is defined as:

$$O = O^s + \mathrm{Proj}(O^l). \quad (6)$$

where $\mathrm{Proj}$ is a learnable linear transformation $\mathbb{R}^d \to \mathbb{R}^d$. Applying this projection to $O^l$ helps reduce the distribution mismatch between softmax and linear attention. Its computational cost is $\mathcal{O}(Nd^2)$, the same as computing $O^l$ and negligible compared with the $\mathcal{O}(N^2d)$ cost of full attention. Specifically, $\mathcal{O}(Nd^2) = 0.004 \times \mathcal{O}(N^2d)$, when $N = 32K, d = 128$ in the Wan2.1-1.3B. In this case, 95% sparsity in the sparse attention component means 94.7% attention complexity reduction.

**Insight.** Linear attention in SLA does not approximate the output corresponding to marginal attention weights, but serves as a learnable compensation that enhances the effectiveness of sparse attention. This is because linear attention alone struggles to approximate the output of full attention (Choromanski et al., 2020; Qin et al., 2022). Therefore, we need to fine-tune the parameters of the target model, enabling it to adapt to the use of linear attention.

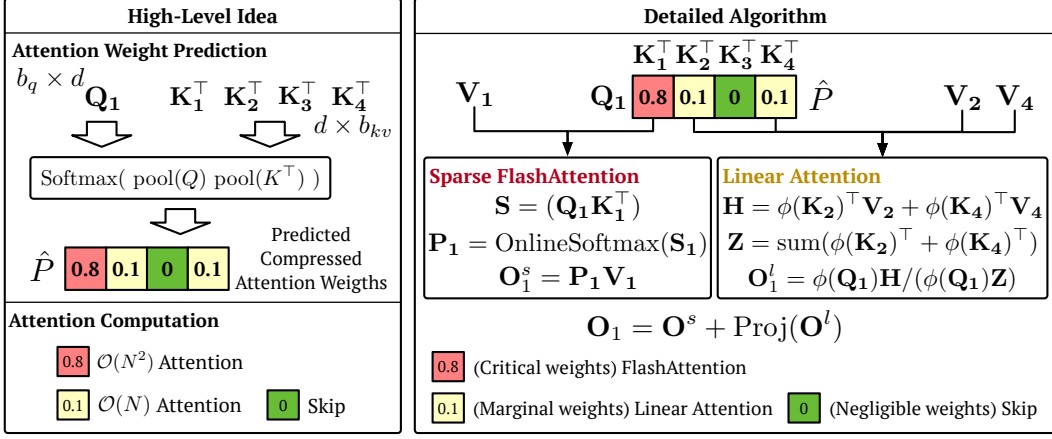

Figure 4: Overview of SLA. The left figure illustrates the high-level idea: attention weights are classified into three categories and assigned to computations of different complexity. The right figure shows the detailed forward algorithm of SLA using the predicted compressed attention weights.

## 5 FINE-TUNING USING SLA

To apply SLA to a diffusion model, we can simply replace the original attention with SLA and fine-tune the model for a few steps on a dataset consistent with the pretraining data. In this section, we describe the forward and backward passes of SLA. Moreover, we detail some additional efficiency optimization for SLA in Appendix A.6.

---

**Algorithm 1:** Forward pass of `SLA`.

---

1: **Input:** Matrices $Q, K, V, Q^\phi, K^\phi \in \mathbb{R}^{N \times d}$, block sizes $b_q, b_{kv}$, hyper-parameters $k_h, k_l$.
2: Divide $Q, Q^\phi$ to $T_m = N/b_q$ blocks $\{\mathbf{Q}_i\}$ and $\{\mathbf{Q}_i^\phi\}$ ;
3: Divide $K, V, K^\phi$ to $T_n = N/b_{kv}$ blocks $\{\mathbf{K}_i\}$, $\{\mathbf{V}_i\}$ and $\{\mathbf{K}_i^\phi\}$ ;
4: $h = \{h_j\} = \{(\mathbf{K}_j^\phi)^\top \mathbf{V}_j\}$ ; $z = \{z_j\} = \{\mathrm{rowsum}((\mathbf{K}_j^\phi)^\top)\}$ ;    // Precompute for linear attention
5: $P_c = \mathrm{Softmax}(\mathrm{pool}(Q)\mathrm{pool}(K)^\top/\sqrt{d})$ ;    Initialize $M_c = 0$ ;
6: $M_c[i, j] = 1$ if $P_c[i, j] \in \mathtt{TopK}(P_c[i, :], k_h)$ ; $M_c[i, j] = 0$ if $P_c[i, j] \in \mathtt{BottomK}(P_c[i, :], k_l)$ ;
7: **for** $i = 1$ **to** $T_m$ **do**
8:     **for** $j = 1$ **to** $T_n$ **do**
9:         **if** $M_c[i, j] = 1$ **then**
10:             $\mathbf{S}_{ij} = \mathbf{Q}_i\mathbf{K}_j^\top/\sqrt{d}$ ;     $m_{ij} = \max(m_{i,j-1}, \mathrm{rowmax}(\mathbf{S}_{ij}))$ ; $\mathbf{P}_{ij} = \exp(\mathbf{S}_{ij} - m_{ij})$ ;
11:             $l_{ij} = e^{m_{i,j-1} - m_{ij}} l_{i,j-1} + \mathrm{rowsum}(\mathbf{P}_{ij})$ ;     $\mathbf{O}_{ij}^s = \mathrm{diag}(e^{m_{i,j-1} - m_{ij}})\mathbf{O}_{i,j-1}^s + \mathbf{P}_{ij}\mathbf{V}_j$ ;
12:         **else if** $M_c[i, j] = 0$ **then**
13:             $\mathbf{H}_i \leftarrow \mathbf{H}_i + h_j$;     $\mathbf{Z}_i \leftarrow \mathbf{Z}_i + z_j$ ;
14:         **end if**
15:     **end for**
16:     $\mathbf{O}_i^s = \mathrm{diag}(l_i^{T_n})^{-1}\mathbf{O}_{i,T_n}^s$ ;   $\mathbf{O}_i^l = \mathbf{Q}_i^\phi\mathbf{H}_i/(\mathbf{Q}_i^\phi\mathbf{Z}_i)$ ;   $\mathbf{L}_i = m_{i,T_n} + \log(l_{i,T_n})$ ;
17: **end for**
18: **return** $O^s = \{\mathbf{O}_i^s\}$, $O^l = \{\mathbf{O}_i^l\}$ ;

---

## 5.1 FORWARD PASS

The formulation of the forward computation was introduced in Section 4. The complete algorithm of the forward pass of `SLA` is presented in Algorithm 1. It's worth noting that we precompute $h_j = \phi(\mathbf{K}_j)^\top\mathbf{V}_j$ and $z_j = \mathrm{rowsum}(\phi(\mathbf{K}_j)^\top)$ for each pair $(K_j, V_j)$ (Line 4 in Algorithm 1). This design ensures that, when $M_c[i, j] = 0$, the corresponding operation only involves a single matrix addition (Line 13 in Algorithm 1), thereby improving efficiency. To simplify the notation, we denote $Q^\phi = \phi(Q)$ and $K^\phi = \phi(K)$ in the following.

---

**Algorithm 2:** Backward pass of `SLA`.

---

1: **Input:** $Q, K, V, Q^\phi, K^\phi, M_c, \{\mathbf{L}_i\}, \{\mathbf{H}_i\}, \{\mathbf{Z}_i\}, O^s, O^l$ from the forward, $dO^s, dO^l \in \mathbb{R}^{N \times d}$.
2: $D^s = \mathrm{rowsum}(dO^s \odot O^s)$, $D^l = \mathrm{rowsum}(dO^l \odot O^l)$, divide $D^s, D^l$ into $T_m$ blocks $\{\mathbf{D}_i^s\}, \{\mathbf{D}_i^l\}$ ;
3: **for** $i = 1$ **to** $T_m$ **do**
4:     $\mathbf{dH}_i = (\mathbf{Q}_i^\phi/(\mathbf{Q}_i^\phi\mathbf{Z}_i))^\top\mathbf{dO}_i^l$;   $\mathbf{dZ}_i = -(\mathbf{Q}_i^\phi/(\mathbf{Q}_i^\phi\mathbf{Z}_i))^\top D_i^l$ ;
5:     $\mathbf{dQ}_i^\phi = (\mathbf{dO}_i^l(\mathbf{H}_i)^\top - \mathbf{D}_i^l\mathbf{Z}_i^\top)/(\mathbf{Q}_i^\phi\mathbf{Z}_i)$ ;
6: **end for**
7: **for** $j = 1$ **to** $T_n$ **do**
8:     Initialize $\mathbf{dH} = 0, \mathbf{dZ} = 0$ ;
9:     **for** $i = 1$ **to** $T_m$ **do**
10:         **if** $M_c[i, j] = 1$ **then**
11:             $\mathbf{S}_{ij} = \mathbf{Q}_i\mathbf{K}_j^\top/\sqrt{d}$ ;  $\mathbf{P}_{ij} = \exp(\mathbf{S}_{ij} - \mathbf{L}_i)$ ;  $\mathbf{dV}_j \leftarrow \mathbf{dV}_j + \mathbf{P}_{ij}^\top\mathbf{dO}_i^s$ ;  $\mathbf{dP}_{ij} = \mathbf{dO}_{ij}^s\mathbf{V}_j^\top$ ;
12:             $\mathbf{dS}_{ij} = \mathbf{P}_{ij} \odot (\mathbf{dP}_{ij} - \mathbf{D}_i^s)$ ;  $\mathbf{dQ}_i \leftarrow \mathbf{dQ}_i + \mathbf{dS}_{ij}\mathbf{K}_j$ ;     $\mathbf{dK}_j \leftarrow \mathbf{dK}_j + \mathbf{dS}_{ij}^\top\mathbf{Q}_i$ ;
13:         **else if** $M_c[i, j] = 0$ **then**
14:             $\mathbf{dH} \leftarrow \mathbf{dH} + \mathbf{dH}_i$;     $\mathbf{dZ} \leftarrow \mathbf{dZ} + \mathbf{dZ}_i$ ;
15:         **end if**
16:     **end for**
17:     $\mathbf{dK}_j^\phi = \mathbf{V}_j(\mathbf{dH})^\top + (\mathbf{dZ})^\top$;     $\mathbf{dV}_j = \mathbf{K}_j^\phi\mathbf{dH}$ ;
18: **end for**
19: **return** $dQ = \{\mathbf{dQ}_i\}$, $dK = \{\mathbf{dK}_i\}$, $dV = \{\mathbf{dV}_i\}$, $dQ^\phi = \{\mathbf{dQ}_i^\phi\}$, $dK^\phi = \{\mathbf{dK}_i^\phi\}$ ;

---

## 5.2 BACKWARD PASS

The backward pass computes gradients for both the sparse and linear components, which are also fused into a single GPU kernel for efficiency.

**Gradient notation.** The prefix $d$ or $\mathbf{d}$ is used to denote gradients, e.g., $dO^s, dO^l$ are the gradients of $O^s, O^l$ with respect to some loss function $\ell$, respectively.

**Sparse attention gradients.** The output gradient $dO^s$ is backpropagated to compute $dQ$, $dK$, and $dV$, following the same derivation as in FlashAttention (Dao, 2023). Given $dO^s$, the backward pass is carried out as follows:

$$\mathbf{dP}_{ij} = \mathbf{dO}_{ij}^s \mathbf{V}_j^\top, \quad \mathbf{D}_i^s = \text{rowsum}(\mathbf{dO}_i^s \odot \mathbf{O}_i^s), \quad \mathbf{dS}_{ij} = \mathbf{P}_{ij} \odot (\mathbf{dP}_{ij} - \mathbf{D}_i^s),$$
$$\mathbf{dQ}_i = \mathbf{dS}_{ij}\mathbf{K}_j, \quad \mathbf{dK}_j = \mathbf{dS}_{ij}^\top \mathbf{Q}_i, \quad \mathbf{dV}_j = \mathbf{P}_{ij}^\top \mathbf{dO}_i^s. \tag{7}$$

Here, we consider $\mathbf{D}_i^s \in \mathbb{R}^{b_q \times 1}$ as a column vector.

**Linear attention gradients.** The gradient $dO^l$ yields $dQ^\phi, dK^\phi, dV$ through the chain rule:

$$\mathbf{dH}_i = \left(\frac{\mathbf{Q}_i^\phi}{\mathbf{Q}_i^\phi \mathbf{Z}_i}\right)^\top \mathbf{dO}_i^l, \quad \mathbf{D}_i^l = \text{rowsum}(\mathbf{dO}_i^l \odot \mathbf{O}_i^l), \quad \mathbf{dZ}_i = -\left(\frac{\mathbf{Q}_i^\phi}{\mathbf{Q}_i^\phi \mathbf{Z}_i}\right)^\top \mathbf{D}_i^l$$
$$\mathbf{dQ}_i^\phi = \frac{(\mathbf{dO}_i^l (\mathbf{H}_i)^\top - \mathbf{D}_i^l \mathbf{Z}_i^\top)}{\mathbf{Q}_i^\phi \mathbf{Z}_i}, \quad \mathbf{dK}_j^\phi = \mathbf{V}_j (\mathbf{dH}_i)^\top + (\mathbf{dZ}_i)^\top, \quad \mathbf{dV}_j = \mathbf{K}_j^\phi \mathbf{dH}_i \tag{8}$$

Here, $\mathbf{dK}_j^\phi$ and $\mathbf{dV}_j$ are obtained by aggregating $\mathbf{dH}_i$ and $\mathbf{dZ}_i$. Similar to the forward pass, each $\mathbf{dH}_i$ and $\mathbf{dZ}_i$ is precomputed so that the remaining computation reduces to simple matrix additions. The detailed algorithm is provided in Algorithm 2.

# 6 EXPERIMENT

## 6.1 SETUP

**Model and Datasets.** We use the Wan2.1-1.3B model (Wan et al., 2025) for video generation experiments in the main text and LightningDiT (Yao et al., 2025) for image generation experiments in the Appendix A.2. We also conduct experiments on a private MM-DiT (Esser et al., 2024) model in the Appendix A.4. For video experiments, we use a private dataset collected from websites such as Pexels (Pexels) and Common Crawl (Common Crawl), consisting of 20,000 5-second videos at 480p resolution for fine-tuning. For image experiments, following LightningDiT (Yao et al., 2025), we use the ImageNet (Deng et al., 2009) dataset at a resolution of $512 \times 512$.

**Baselines.** We compare `SLA` with state-of-the-art sparse attention methods applicable to diffusion models, including (1) `VSA` (Zhang et al., 2025e), (2) `VMoBa` (Wu et al., 2025), and (3) the training-free SparseAttn (Zhang et al., 2025b) (`Sparge-F`) and (4) a trainable implementation of SpargeAttn (`Sparge-T`). For `VSA` and `VMoBa`, we use their official implementations, while for `Sparse-T`, we implement the method ourselves because there is no official implementation. In addition, we design several baselines for ablation studies: (5) `Linear Only`, which applies only linear attention; (6) `Sparse Only`, which applies only the sparse attention component of `SLA`; and (7) `L+S`, which directly sums the attention outputs of the `Linear Only` and `Sparse Only`.

**Metrics.** For video quality, following Zhang et al. (2024a); Yang et al. (2025b), we use four evaluation dimensions of VBench (Zhang et al., 2024a): Imaging Quality (`IQ`), Overall Consistency (`OC`), Aesthetic Quality (`AQ`), Subject Consistency (`SC`). We also use the Vision Reward (`VR`) (Xu et al., 2024) for human preference evaluation, Aesthetic Video Quality (`VA`), and Technical Video Quality (`VT`) (Liu et al., 2023). For image quality, following Yao et al. (2025), we use `FID`. For attention computation complexity, we use `FLOPs` (floating point of operations). For attention efficiency, we use `FLOPS` (floating-point operations per second) for attention kernel efficiency. Specifically, `FLOPS` here is $\mathcal{O}(\text{full attention})/t$, where $\mathcal{O}(\cdot)$ denotes the operation count and $t$ the attention latency. We use seconds for end-to-end generation latency.

**Hyper-parameters.** We use a training batch size of 64 and fine-tune the Wan2.1 model for 2000 steps. For the activation function $\phi$, we use softmax according to our ablation experiments. $k_h\%$ is 5% and $k_l\%$ is 10%. For block size, we use $b_q = b_{kv} = 64$. The hyper-parameters for image generation tasks are detailed in Appendix A.2.

## 6.2 EFFECTIVENESS

Table 1 compares the video generation quality and efficiency of `SLA` with baseline methods on Wan2.1-1.3B, fine-tuned separately with `SLA`, Full Attention, and each baseline. `SLA` delivers about

a $19.3\times$ efficiency gain while maintaining video quality comparable to Full Attention. Moreover, compared with the baselines, SLA consistently achieves higher quality even under greater sparsity. For example, 95% (1-**5%**) sparsity in SLA is actually about $3\times$ more efficient than 85% (1-**15%**) while still producing better video quality.

Table 1: Quality and efficiency comparison of SLA and other baseline methods.

| Method | Quality | | | | | | | Efficiency | |
|---|---|---|---|---|---|---|---|---|---|
| | VA ↑ | VT ↑ | IQ ↑ | OC ↑ | AQ ↑ | SC ↑ | VR ↑ | FLOPs ↓ | Sparsity ↑ |
| Full Attention | 76.78 | 82.88 | 62.5 | 23.3 | 56.1 | 93.0 | 0.059 | 52.75T | 0% |
| Sparge-F | 0.002 | 0.026 | 26.0 | 4.6 | 35.7 | 85.1 | -0.216 | 7.91T | 85% |
| Sparge-T | 73.83 | 77.87 | 61.9 | 22.7 | 55.4 | 93.1 | 0.014 | 7.38T | 84% |
| VMoBa | 32.33 | 35.79 | 58.0 | 18.8 | 46.2 | 89.9 | -0.175 | 7.91T | 85% |
| VSA | 55.37 | 64.61 | 60.6 | 22.4 | 51.9 | 83.6 | -0.069 | 5.92T | 89% |
| SLA | **76.96** | **83.92** | **62.2** | **23.6** | **55.9** | 93.1 | **0.048** | **2.74T** | **95%** |

Table 2: Ablation results for SLA.

| Method | Quality | | | | | | | Efficiency | |
|---|---|---|---|---|---|---|---|---|---|
| | VA ↑ | VT ↑ | IQ ↑ | OC ↑ | AQ ↑ | SC ↑ | VR ↑ | FLOPs ↓ | Sparsity ↑ |
| Full Attention | 76.78 | 82.88 | 62.5 | 23.3 | 56.1 | 93.0 | 0.059 | 52.75T | 0% |
| Linear Only | 0.042 | 0.099 | 39.5 | 3.6 | 28.8 | 90.7 | -0.213 | 0.10T | 100% |
| Sparse Only | 64.00 | 70.50 | 57.2 | 21.8 | 51.7 | 88.7 | -0.073 | 7.91T | 85% |
| L+S | 29.65 | 41.15 | 58.6 | 18.8 | 45.3 | 87.1 | -0.105 | 5.37T | 90% |
| SLA (softmax) | **76.96** | **83.92** | 62.2 | **23.6** | **55.9** | 93.1 | 0.048 | **2.73T** | **95%** |
| SLA (elu+1) | 75.50 | 81.01 | 62.8 | 23.5 | 55.3 | 92.9 | 0.034 | 2.74T | 95% |
| SLA (hedgehog) | 74.59 | 82.62 | 61.9 | 22.5 | 54.3 | 93.2 | 0.035 | 3.11T | 95% |
| SLA (Top 5%) | 76.96 | 83.92 | 62.2 | 23.6 | 55.9 | 93.1 | 0.048 | 2.73T | 95% |
| SLA (Top 10%) | 75.29 | 82.20 | 62.5 | 22.6 | 55.8 | 93.5 | 0.057 | 5.38T | 90% |
| SLA (Top 20%) | 75.81 | 83.82 | 62.7 | 22.4 | 54.5 | 92.6 | 0.059 | 10.65T | 80% |

Table 3: Quality of SLA in the zero-shot and limited finetuning settings.

| Method | Quality | | | | | | |
|---|---|---|---|---|---|---|---|
| | VA ↑ | VT ↑ | IQ ↑ | OC ↑ | AQ ↑ | SC ↑ | VR ↑ |
| Full Attention | 76.78 | 82.88 | 62.5 | 23.3 | 56.1 | 93.0 | 0.059 |
| SLA (0 step) | 41.11 | 51.79 | 58.3 | 21.4 | 46.7 | 81.0 | -0.1105 |
| SLA (250 steps) | 64.46 | 78.06 | 59.0 | 22.8 | 55.7 | 88.5 | -0.0244 |
| SLA (1000 steps) | 74.58 | 80.09 | 61.8 | **23.7** | **56.1** | 92.3 | 0.0429 |
| SLA (2000 steps) | **76.96** | **83.92** | **62.2** | 23.6 | 55.9 | **93.1** | **0.0483** |

## 6.3 EFFICIENCY

Figure 6 compares the kernel speed and end-to-end latency of SLA on Wan2.1–1.3B with an RTX5090. Note that even VSA in 89% sparsity and VMoBa in 85% sparsity, their generation quality is already worse than SLA, so higher sparsity settings (e.g., 95%) are not quality-matched comparisons. In the forward pass, SLA achieves a $13.7\times$ speedup over FlashAttention2 and is $1.93\times$ faster than VSA with 95% sparsity and $3.36\times$ faster than VMoBa with 95% sparsity. In the backward pass, it delivers a $6.8\times$ speedup over FlashAttention2, still outperforming VSA and VMoBa. For end-to-end video generation, SLA reduces attention latency from 97s to 11s ($8.8\times$ reduction), resulting in a $2.2\times$ end-to-end speedup. For fine-tuning overhead, we train Wan2.1-1.3B for only 2,000 steps with a batch size of 64, which is less than 0.1% of the cost of pretraining (typically $10^5$–$10^6$ steps with a batch size of $10^3$–$10^4$) (Wan et al., 2025). The finetuning of SLA requires approximately 9 hours on 8 NVIDIA H200 GPUs.

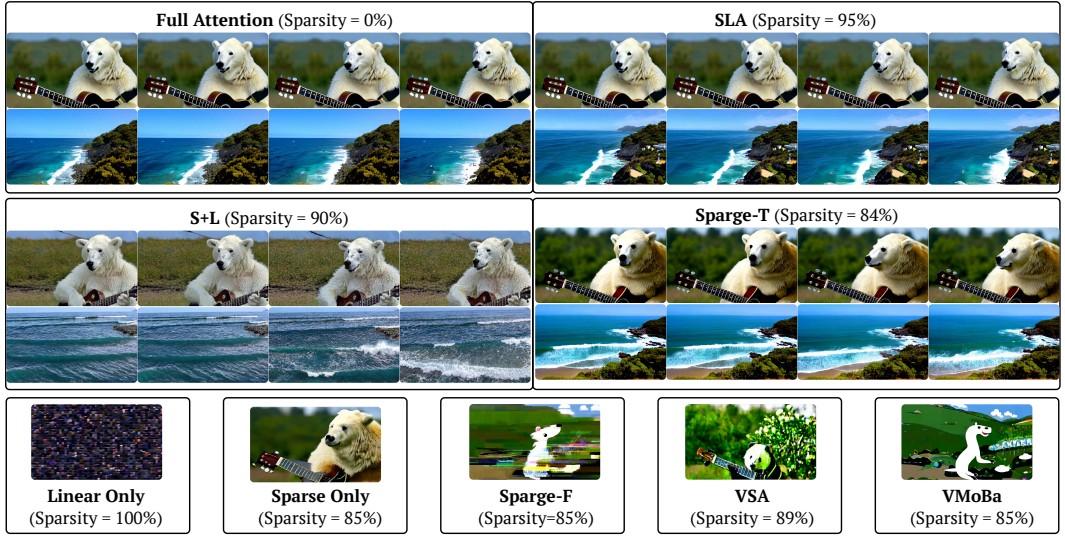

Figure 5: Video examples using Wan2.1 fine-tuned with `SLA` and baselines. For `Linear Only`, `Sparse Only`, `Sparge-F`, `VSA`, and `VMoBa`, only a single frame per prompt is shown, as their video quality is not sufficient. The full visible comparison is in Figure 7 in Appendix A.1.

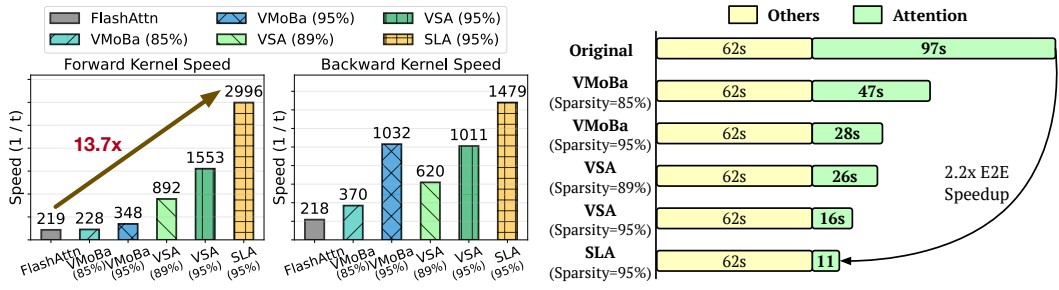

(a) Attention GPU Kernel speed comparison on RTX5090.       (b) End-to-end video generation latency comparison.

Figure 6: Attention kernel speed and end-to-end generation latency of `SLA` and baselines on Wan2.1-1.3B with RTX5090. FlashAttn refers to FlashAttn2, the fastest available version on RTX5090.

In Appendix A.7, we compare the efficiency of `SLA` and FlashAttention on more GPUs, while in Appendix A.9, we explore LoRA (Hu et al., 2022) as a more efficient finetuning paradigm.

## 6.4 ABLATION STUDY

**Fusing sparse and linear attention.** To evaluate the effectiveness of `SLA` in integrating sparse and linear attention, we compare `SLA` with `Sparse Only`, `Linear Only`, and `S+L` on Wan2.1 in terms of end-to-end generation quality and efficiency. As shown in Table 2, `SLA` achieves the best generation quality and is more efficient than `Sparse Only` and `S+L`, confirming the effectiveness of our fusion strategy.

**Activation function in linear attention.** To study the effect of the activation function $\phi$ in the linear attention component of `SLA`, we evaluate softmax, elu+1, and hedgehog. Results in Table 2 show that softmax generally provides better generation quality and efficiency.

**Impact of parameter $k_h$.** We vary $k_h$ from 5% to 20% and report the results in Table 2. We find that $k_h = 5\%$ already yields generation quality close to that of full attention. Since $k_h = 5\%$ saves about half and a quarter of the computation compared with $k_h = 10\%$ and $k_h = 20\%$, it offers the best trade-off between efficiency and quality.

**Impact of parameters $k_l, b_q$ and $b_{kv}$.** These parameters appear to have a smaller influence compared with $k_h$. We conduct ablation experiments on the LightningDiT model, and the results are reported in Appendix A.3.

**Zero-shot and limited finetuning.** Table 3 presents the performance of `SLA` under zero-shot (0-step finetuning) and limited finetuning budgets. The quality of videos generated by `SLA` steadily improves as finetuning progresses. After only 1K finetuning steps, the quality is already close to that of full attention, and 2K steps yield the best results.

## 6.5 VISIBLE EXAMPLES

Figure 5 and Figure 7 show video examples from Wan2.1-1.3B fine-tuned using `SLA` and baselines. `SLA` produces videos comparable to full attention even at **95**% sparsity, while other methods exhibit noticeable distortions even at sparsity levels below 90%.

## 7 RELATED WORK

As sequence lengths in generative models (e.g., language and video) grow, the quadratic cost of attention becomes a key bottleneck. Many studies aim to improve efficiency in two main directions: sparse and linear attention. Most sparse attention methods (Xiao et al., 2024b;a; Jiang et al., 2024; Gao et al., 2024; Fu et al., 2024; Xi et al., 2025; Zhang et al., 2025b; Ribar et al., 2023; Yang et al., 2025a; Xi et al., 2026) speed up inference without training by masking computation at test time. Some (Zhang et al., 2025e; Wu et al., 2025; Zhang et al., 2026b;c) add sparsity during training, enabling higher sparsity. Linear attention methods (Wang et al., 2020; Choromanski et al., 2020; Katharopoulos et al., 2020; Qin et al., 2024; Yang et al., 2024; Sun et al., 2023) are mainly studied in language models. For DiT, SANA (Xie et al., 2024) and Dig (Zhu et al., 2025) show linear attention works for image generation pre-training, but in video generation, existing methods cannot rely on it alone for lossless quality. Another direction is hardware-efficient attention (Dao et al., 2022; Dao, 2023; Shah et al., 2024; Zhang et al., 2024c;b; 2025c;a), which optimizes GPU execution through tiling, kernel fusion, and quantization.

## 8 CONCLUSION

We propose `SLA`, a trainable attention that unifies sparse and linear attention to accelerate Diffusion Transformers. `SLA` assigns computation according to importance: it computes $\mathcal{O}(N^2)$ attention for critical weights, $\mathcal{O}(N)$ attention for marginal weights, and skips negligible computations. This design enables substantial reductions in attention cost while preserving effectiveness. Experiments show that just a few fine-tuning steps enable `SLA` to accelerate models effectively. Specifically, `SLA` achieves about **20**× reduction in attention computation, along with a **13**.**7**× GPU kernel speedup and a **2**.**2**× end-to-end speedup on Wan2.1-1.3B, all without degrading the quality of video generation.

## ETHICS STATEMENT

This work proposes a method for improving the efficiency of Diffusion Transformers. The study does not involve human subjects, personally identifiable information, or sensitive data. We believe the proposed method does not raise ethical concerns beyond standard considerations in efficient model design.

## ACKNOWLEDGEMENTS

This work was supported by the Fundamental and Interdisciplinary Disciplines Breakthrough Plan of the Ministry of Education of China (No. JYB2025XDXM101); the NSFC Projects (Nos. 62550004, 92270001, 62376131). J.Z is also supported by the XPlorer Prize.

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

# A APPENDIX

## A.1 MORE VISIBLE EXAMPLES

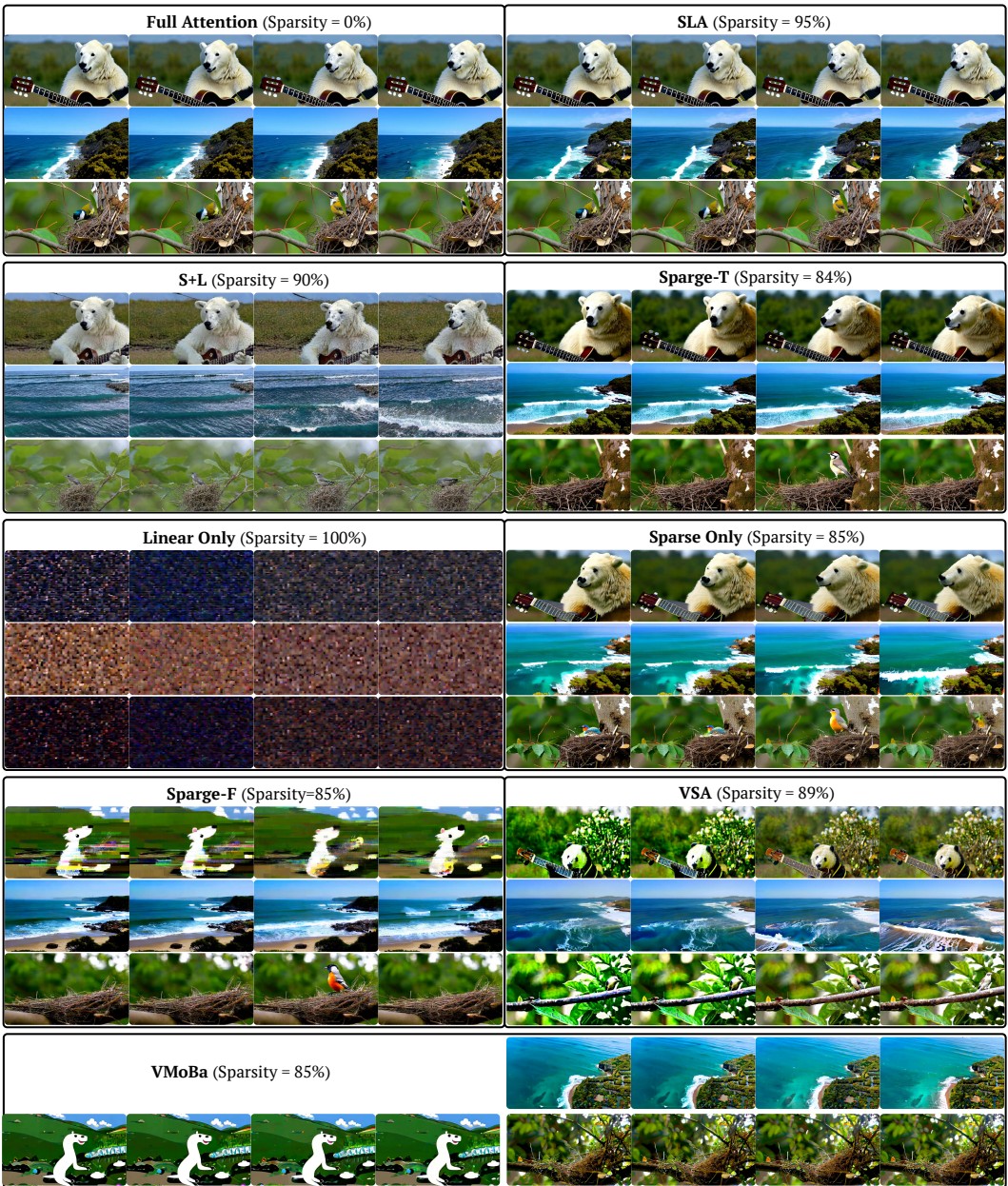

Figure 7: Full video examples generated by the Wan2.1 fine-tuned with SLA and baseline methods. The first prompt is *"A polar bear is playing guitar"*. The second prompt is *"Pacific coast, carmel by the sea ocean and waves"*. The third prompt is *"a bird building a nest from twigs and leaves"*.

Figure 7 demonstrates some additional video examples generated by the Wan2.1 model, fine-tuned with SLA and other attention methods. We can find that SLA consistently achieves higher quality even under bigger sparsity than baselines.

## A.2 EXPERIMENTS FOR IMAGE GENERATION

**Experimental setup.** As described in Section 6.1, we evaluate `SLA` and the baselines on a pretraining task of LightningDiT (Yao et al., 2025). Specifically, we use the `LightningDiT-1p0B/1` model, consisting of 1.03B parameters, trained on the ImageNet (Deng et al., 2009) dataset at a resolution of $512 \times 512$. We introduce a new baseline block diagonal as a representative of static pattern methods. It splits the sequence into blocks and computes attention only within each block.

**Hyperparameters.** All hyperparameters follow (Yao et al., 2025), except that we train for 100000 steps with a batch size of 128. For `SLA`, we set $\phi$ to softmax and use a block size of $b_q = b_{kv} = 64$.

**Metrics.** Following (Yao et al., 2025), we adopt `FID` to assess image quality and FLOPs to measure computational complexity.

**Results.** The results are summarized in Table 4. At the highest sparsity level, `SLA` outperforms all other baselines and even surpasses full attention on the `FID` metric, confirming the advantage of `SLA` in preserving image quality. This finding is consistent with the video experiments on Wan2.1 reported in Section 6.2.

Table 4: Quality and efficiency comparison of `SLA` and other baselines on image generation.

| Method | Quality | Efficiency | |
|---|---|---|---|
| | FID $\downarrow$ | FLOPs $\downarrow$ | Sparsity $\uparrow$ |
| Full Attention | 31.87 | 12.88G | 0% |
| SpargeAttn-F | 206.11 | 3.66G | 71.57% |
| SpargeAttn-T | 46.05 | 3.16G | 75.45% |
| VSA(2D) | 35.75 | 3.62G | 75.00% |
| VMoBA(2D) | 39.45 | 3.22G | 75.00% |
| SLA | **31.49** | **1.73G** | **87.50%** |
| Block Diagonal | 55.06 | 3.22G | 75.00% |

## A.3 ABLATION STUDY ON IMAGE GENERATION

In addition to the ablation study conducted on the Wan2.1-1.3B model in Section 6.4, we further investigate the impact of the parameters $k_l$, $b_q$ and $b_{kv}$ on the LightningDiT model. Table 5 shows that varying $b_q$ and $b_k$ leads to only marginal changes in performance, while increasing $k_l$ consistently degrades generation quality.

Table 5: Ablation Study on $k_l, b_q$ and $b_{kv}$.

| $b_q$ | $b_{kv}$ | $k_l$ | FID $\downarrow$ |
|---|---|---|---|
| 64 | 64 | 0% | 31.49 |
| 128 | 64 | 0% | 31.49 |
| 128 | 32 | 0% | 31.74 |
| 64 | 64 | 6.25% | 32.94 |
| 64 | 64 | 12.5% | 34.04 |
| 64 | 64 | 25% | 37.31 |

## A.4 EXPERIMENTS ON MM-DiT MODEL

**Experimental setup.** To demonstrate that `SLA` can be applied to models with different attention layouts, we evaluate `SLA` and the baseline on a finetuning task using a private MM-DiT model.

**Hyperparameters.** All hyperparameters are the same as those in Section 6.1.

**Metrics.** Following Section 6.1, we use `VA`, `IQ`, `OC`, `AQ`, and `SC` to measure video quality.

**Results.** The results are summarized in Table 6. `SLA` has almost no loss in end-to-end metrics compared to full attention, indicating that `SLA` generalizes well to MM-DiT architectures.

Table 6: Quality comparison of `SLA` and baseline on MM-DiT model.

| Method | Quality | | | | |
|---|---|---|---|---|---|
| | VA ↑ | IQ ↑ | OC ↑ | AQ ↑ | SC ↑ |
| Full Attention | 57.81 | 0.58 | 7.88 | 55.77 | 86.89 |
| SLA (ours) | 55.18 | 55.97 | 8.28 | 56.11 | 88.50 |

## A.5 EXPERIMENTS USING TOP-CDF

Because top-CDF (Zhang et al., 2025b) is a common alternative for top-k when forming the sparse mask $M_c$, particularly to improve robustness against the inaccuracies in $P_c$, we compare the two approaches in Table 7. Our results demonstrate that top-k consistently delivers better video quality, suggesting that `SLA` is not overly sensitive to small errors in $P_c$.

Table 7: Experiments of combining `SLA` and additional tricks.

| Method | Quality | | | | |
|---|---|---|---|---|---|
| | IQ ↑ | OC ↑ | AQ ↑ | SC ↑ | VR ↑ |
| Full Attention | 62.5 | 23.3 | 56.1 | 93.0 | 0.059 |
| SLA (Top-k) | **62.2** | **23.6** | **55.9** | **93.1** | **0.0483** |
| SLA (Top-CDF) | 60.8 | 22.2 | 53.8 | 91.2 | -0.0263 |

## A.6 ADDITIONAL EFFICIENCY OPTIMIZATION

Since the efficiency of `SLA` depends heavily on the sparsity pattern, we introduce several complementary optimizations tailored to different sparsity levels. These optimizations lead to substantial gains in computational efficiency:

**Lookup table.** When $M_c$ is highly sparse (e.g., sparsity $> 90\%$), scanning entire rows or columns to read mask values causes significant memory overhead. To mitigate this, we preprocess the nonzero positions of each row and column and store them in a lookup table. During computation, only the lookup table is accessed, substantially reducing memory traffic.

**Pre-aggregation for linear attention.** Although Line 13 in Algorithms 1 and Line 14 in Algorithm 2 require only a single matrix addition, repeatedly performing such additions incurs high overhead when many entries of $M_c$ are 0 (e.g., $> 90\%$). To address this, we precompute the row/column sums $\sum_j h_j$ and $\sum_j z_j$, and then subtract the contributions corresponding to $M_c[i, j] \neq 0$. In this way, 90% of the additions can be replaced by only 10% subtractions.

**Method of Four Russians.** When the number of blocks with $M_c[i, j] = 0$ is neither very small nor very large (e.g., around 50%), we provide an efficient implementation for Line 13 in Algorithms 1 and Line 14 in Algorithm 2. Specifically, we adopt the Method of Four Russians (Arlazarov et al., 1970). The key idea is to group $h_j$ and $z_j$ into segments of $g$ consecutive blocks and precompute all $2^g$ possible subset sums within each segment. During the forward pass, any subset of $g$ blocks can then be obtained by a single look-up, rather than summing them on the fly. This scheme allows a theoretical computation reduction by $1/g$.

## A.7 EFFICIENCY TEST ON MORE DEVICES

While `SLA` achieves substantial speedups over FlashAttention on the RTX5090, we further evaluate its performance across a broader range of GPUs to demonstrate its generality. Table 8 reports the speedup of `SLA` over FlashAttention on RTX4090, A800, L20, and H800 at $95\%$ sparsity. Across all

devices, `SLA` delivers at least a 10x acceleration in attention computation, highlighting its consistent efficiency gains across diverse hardware.

Table 8: Speedup of `SLA` on more devices.

| GPU | Speedup |
|---|---|
| RTX4090 | 11.20x |
| A800 | 10.76x |
| L20 | 13.18x |
| H800 | 12.94x |

## A.8 MEMORY FOOTPRINTS OF `SLA`

To provide more hardware details for the `SLA` kernels, we report the memory footprints of `SLA` at $k_h = 10\%$ in Table 9, including MFU, L2 cache I/O utilization, DRAM I/O utilization, and peak memory usage relative to FlashAttention for both the forward and backward passes. The profiling results indicate that `SLA` is a compute-bound workload, with no evident memory bottlenecks.

Table 9: Memory footprints of `SLA`.

| Kernel | MFU | L2 Cache Util. | DRAM Util. | Memory Usage |
|---|---|---|---|---|
| `SLA` Foward | 75.5% | 63.8% | 2.9% | 1.12x |
| `SLA` Backward | 54.2% | 57.4% | 1.8% | 1.15x |

## A.9 COMBINING `SLA` WITH LoRA

We evaluate the combination of `SLA` with LoRA ($r = 8, \alpha = 4$) to explore more efficient fine-tuning strategies. The results, summarized in Table 10, show there is a clear performance gap between the LoRA setup and full-parameter finetuning.

Although more aggressive hyperparameter tuning (e.g., larger ranks) might reduce this gap, the current results indicate that full finetuning is a more reliable way for `SLA`.

Table 10: Experiments of combining `SLA` with LoRA.

| Method | Quality | | | |
|---|---|---|---|---|
| | IQ ↑ | OC ↑ | AQ ↑ | SC ↑ |
| Full Attention | 62.5 | 23.3 | 56.1 | 93.0 |
| `SLA` | **62.2** | **23.6** | **55.9** | **93.1** |
| `SLA` + LoRA | 85.0 | 56.3 | 51.8 | 22.9 |

## USE OF LARGE LANGUAGE MODELS

We used a language model only for polishing English writing, while all ideas, experiments, results, and interpretations are our own.

