# OpenReview forum: "SLA: Beyond Sparsity in Diffusion Transformers via Fine-Tunable Sparse–Linear Attention"
_ICLR.cc/2026/Conference — ICLR 2026 Poster_

### Official Review · Reviewer_24AN · 2025-10-24

**Soundness:** 3
**Presentation:** 3
**Contribution:** 3
**Rating:** 6
**Confidence:** 4

**Summary:**

This paper proposed SLA, a sparse-linear attention for efficient diffusion transformer generation. SLA finds that the attention weights can be decoupled into two types: a small fraction of large weights with high rank and the remaining weights with very low rank. SLA further designed a trainable attention method that fuses sparse and linear attention to accelerate diffusion models. Experiments shown that SLA reduces attention computation by 95% without degrading end-to-end generation quality, with 2.2 times speedup for video generation.

**Strengths:**

1.SLA designed efficient forward and backward propagation modules, which are very complete and valuable in engineering.

2.It is very reasonable and meaningful for SLA to decompose attention into two computing modes, and achieve advanced performance.

3.SLA has great advantages in efficiency and has achieved state-of-the-art acceleration performance.

**Weaknesses:**

1.Is the attention mode mentioned in SLA commonly present in existing advanced video generation models? Reporting more models such as CogVideoX's results can further demonstrate the universality of this phenomenon.

2.How is Eq3 calculated specifically? For example, directly calculating the average value or calculating the rank? If it is the average, is there a direct correlation between this and the rank mentioned before?

3.Does the implementation of SLA distinguish between model architectures, such as Wan2.1 using DiT architecture, while other models such as HunyuanVideo and CogVideoX use MM-DiT architecture? Will this lead to differences in attention calculation methods and implementation?

4.It is meaningful to conduct additional supplementary experiments on other architectures such as CogVideoX-2B based on MM-DiT architecture.

5.The paper should report the resources consumed during training, such as the number of GPUs and time. And exploring more efficient fine-tuning paradigms such as LoRA can further enhance the practicality of SLA.

**Questions:**

Please see above weaknesses.

---

> ### Author Response · Authors · 2025-11-20
> **Authors’ Rebuttal to the Weaknesses Raised by Reviewer 24AN**
>
> Dear Reviewer 24AN,
> Thank you for your valuable suggestions and questions. Below, we address each point raised.
>
> ---
> > **W1.** Is the attention mode mentioned in SLA commonly present in existing advanced video generation models? Reporting more models such as CogVideoX's results can further demonstrate the universality of this phenomenon.
>
> **Reply:** Thank you for the suggestion. This observation appears to be general for diffusion models. To further validate it, we sample many tensors from CogVideoX and compute the ranks of their attention maps, then report the average results in the Table below. After removing the top 8% values from the attention maps, the remaining have very low rank on average, which supports that our observation holds across different video generation models.
>
> ||Full Attention Map|Top 8% Values|Bottom 92% Values|
> |-|-|-|-|
> |Matrices Rank|1876|1905|4|
>
> ---
> > **W2.** How is Eq3 calculated specifically? For example, directly calculating the average value or calculating the rank? If it is the average, is there a direct correlation between this and the rank mentioned before?
>
> **Reply:** We use ranking instead of averaging. Concretely, given the scores from Eq. (3) and a chosen top $k_h$ (for example, $k_h$ = 5%), we sort all scores and then select the positions of the largest $k_h$ values. These selected positions are marked as 1 in the mask, and all others are marked as 0.
>
> ---
> > **W3.** Does the implementation of SLA distinguish between model architectures, such as Wan2.1 using DiT architecture, while other models such as HunyuanVideo and CogVideoX use MM-DiT architecture? Will this lead to differences in attention calculation and implementation?
>
> **Reply:** SLA can be directly applied to MM-DiT, using the same implementation as in standard DiT. In practice, we can feed the joint $Q, K, V$ over both text and video tokens into SLA directly. To show that SLA also works on MM-DiT, we add an experiment on a private MM-DiT model. As shown in the Table below, using SLA with top-k = 5% and 2K fine-tuning steps, the MM-DiT model has almost no loss in end-to-end metrics compared to full attention. These results indicate that SLA generalizes well to MM-DiT architectures.
>
> |Method|VA ↑|IQ ↑|OC ↑|AQ ↑|SC ↑|
> |-|-|-|-|-|-|
> |Full Attention|57.81|58.15|7.88|55.77|86.89|
> |SLA|55.18|55.97|8.28|56.11|88.50|
>
> In addition, there is an even more conservative way to use SLA on MM-DiT. We can decompose the attention computation into four parts: text-to-video, text-to-text, video-to-video, and video-to-text. SLA is then applied only to the video-to-video part, while keeping the other parts in full attention. This usage is almost identical to the standard DiT setting. Moreover, since the number of video tokens is much larger than that of text tokens (typically about $50\text{K}$ vs. $1\text{K}$), the video-to-video attention dominates the attention cost, so accelerating only this part make sense.
>
> ---
> > **W4.** It is meaningful to conduct additional supplementary experiments on other architectures such as CogVideoX-2B based on MM-DiT architecture.
>
> **Reply:** Thank you for the suggestion. To show that SLA also works on MM-DiT, we add an experiment on a private MM-DiT model. As shown in the Table below, using SLA with top-k = 5% and 2K fine-tuning steps, the MM-DiT model has almost no loss in end-to-end metrics compared to full attention.
>
> |Method|VA ↑|IQ ↑|OC ↑|AQ ↑|SC ↑|
> |-|-|-|-|-|-|
> |Full Attention|57.81|58.15|7.88|55.77|86.89|
> |SLA|55.18|55.97|8.28|56.11|88.50|
>
> This experiment confirms that SLA can be applied to MM-DiT, and we will add this experiment in our final paper.
>
> ---
> > **W5.** The paper should report the resources consumed during training, such as the number of GPUs and time. And exploring more efficient fine-tuning paradigms such as LoRA can further enhance the practicality of SLA.
>
> **Reply:** First, for the training resources, the training of SLA requires approximately 9 hours on 8 NVIDIA H200 GPUs. Second, we explore LoRA as an efficient alternative using standard settings ($r=8, \alpha=4$). As shown in the Table below, there is still a clear performance gap between this LoRA setup and full-parameter fine-tuning. Although more aggressive hyperparameter tuning (e.g., larger ranks) might reduce this gap, the current results indicate that full fine-tuning is a more reliable way for SLA. In practice, we find that SLA is easy to use in a model merging way: the parameter difference from full-parameter fine-tuning can be merged into other fine-tuned or distilled models, so the attention acceleration of SLA can be transferred to these models.
>
> |Metric|Full fine-tuning|LoRA ($r=8$)|
> |-|-|-|
> |SC ↑|93.1|85.0|
> |IQ ↑|62.2|56.3|
> |AQ ↑|55.9|51.8|
> |OC ↑|23.6|22.9|
>
> ---
> **If you feel your concerns have been resolved, we would greatly appreciate it if you consider raising the score. We will add the experiment and analysis in the rebuttal to our paper.**

---

### Official Review · Reviewer_PbE1 · 2025-10-29

**Soundness:** 3
**Presentation:** 3
**Contribution:** 3
**Rating:** 6
**Confidence:** 4

**Summary:**

This paper proposes Sparse–Linear Attention (SLA), a trainable hybrid attention mechanism for Diffusion Transformers (DiTs), particularly targeting video generation models. The core observation is that attention weight matrices can be decomposed into a small set of high-rank, high-magnitude entries and a large set of low-rank, low-magnitude entries. SLA classifies block-wise attention weights into critical, marginal, and negligible categories, applying standard sparse FlashAttention to the critical blocks, linear attention to the marginal blocks, and skipping negligible blocks. The authors implement a fused GPU kernel covering both forward and backward passes and demonstrate—on Wan2.1-1.3B video DiT and LightningDiT image models—that SLA achieves up to 95 % sparsity, 13.7× attention kernel speedup, and 2.2× end-to-end generation speedup with minimal quality degradation.

**Strengths:**

* Originality: The sparse/linear decomposition of attention weights for DiTs is a compelling insight. Rather than naively combining sparse and linear attention, SLA’s dynamic block classification and fused kernel design show genuine innovation tailored to diffusion workloads, pushing beyond previous sparse-only or linear-only acceleration schemes.
* Quality: The methodology is carefully engineered: Algorithm 1 and 2 detail the fused forward/backward passes; precomputation, lookup tables, and Method-of-Four-Russians optimizations indicate a strong systems perspective. Experimental evidence spans both video (Wan2.1) and image (LightningDiT) domains, with comprehensive metrics including VBench dimensions, VR, VA/VT, FID, FLOPs, sparsity, and latency.
* Clarity: The paper is generally well written. Figures 1–6 effectively motivate the decomposition, illustrate quality impacts, and benchmark performance. The block-level notation and discussion of online softmax, pooling, and projection clarify the mechanism.
Significance: Attention cost is the dominant bottleneck in DiTs, especially for long video sequences (10K–100K tokens). Demonstrating a 20× reduction in attention FLOPs with negligible quality loss is highly impactful for scalable diffusion-based video generation. The approach could influence both research and production systems.

**Weaknesses:**

* Dependence on fine-tuning: The method relies on retraining/fine-tuning to adapt models to SLA. While 2K steps are claimed to be minor, the paper provides limited analysis of how SLA behaves in zero-shot replacement or under constrained finetuning budgets (e.g., small datasets, limited steps). A sensitivity study on fine-tuning duration and data diversity would strengthen the practical narrative.
* Generalization beyond the tested architectures: Experiments focus on Wan2.1-1.3B and LightningDiT. Both employ similar attention layouts. It remains unclear whether SLA’s block classification and kernel fusion generalize to other DiT variants (e.g., different block sizes, mixed precision, cross-attention layers, text-video architectures). More diversity in model families (or a detailed discussion of expected behavior) would be valuable.
* Ablation breadth: While Table 2 covers fusion, activation, and kh variations, the key hyper-parameter pair (kh, kl) and block size (bq, bkv) are not jointly explored. Similar attention heads with differing dimensions may require different thresholds; without a deeper ablation, reproducibility across setups could be challenging.
* Evaluation scope: The paper reports strong aggregate metrics but lacks qualitative failure analysis. For instance, where does SLA still degrade outputs (e.g., motion consistency over long horizons, rare events)? Additionally, comparisons to other hybrid approaches (if any exist) or to recent structured sparsity methods in diffusion (e.g., block-diagonal or strided patterns) are absent.
* Kernel integration specifics: Although Appendix A.3 sketches optimization tricks, concrete profiling (e.g., breakdown of compute vs. memory bottlenecks, impact on backward pass peak memory) is missing. Readers aiming to reimplement SLA might struggle without more hardware-level detail.

**Questions:**

* Fine-tuning sensitivity: How does SLA perform if fine-tuning resources are restricted (fewer steps, smaller batch, or lower-resolution data)? Could you provide a curve illustrating quality vs. fine-tuning budget to justify the “few steps” claim quantitatively?
* Mask prediction robustness: The compressed mask Mc relies on pooled Q/K projections. How sensitive is SLA to inaccuracies in Pc during inference, especially when encountering distribution shifts (e.g., different video domains)? Have you evaluated adaptive thresholds or confidence measures?
* Backward pass efficiency: The backward kernel reportedly yields a 6.8× speedup, but details are sparse. Can you share profiling numbers on gradient computation vs. memory traffic? Are there cases where the backward pass becomes the new bottleneck?
Model generality: Have you attempted SLA on cross-attention or text-conditioned video generation models? If not, what modifications would be necessary to extend SLA to multi-head, multi-source attention patterns?
* Failure cases: Could you provide examples or statistics of prompts where SLA underperforms full attention (e.g., higher error metrics, visual artifacts)? Understanding the boundaries of applicability would help practitioners adopt the method safely.
* Hyper-parameter tuning: For practitioners, how should kh and kl be selected for a new model? Do you recommend starting from a certain sparsity target (e.g., 95%) and adjusting based on validation loss, or is there a principled heuristic derived from attention statistics?

---

> ### Author Response · Authors · 2025-11-20
> **Authors’ Rebuttal-1 to the Weaknesses Raised by Reviewer PbE1**
>
> Dear Reviewer PbE1,
> Thank you for your valuable suggestions and questions. Below, we address each point raised.
>
> ---
>
> > **W1.** Dependence on fine-tuning: The method relies on retraining/fine-tuning to adapt models to SLA. While 2K steps are claimed to be minor, the paper provides limited analysis of how SLA behaves in zero-shot replacement or under constrained finetuning budgets (e.g., small datasets, limited steps). A sensitivity study on fine-tuning duration and data diversity would strengthen the practical narrative.
>
> **Reply:** Thank you for the valuable suggestion. We add additional experiments to study SLA in the zero-shot and limited fine-tuning settings. As shown in the Table below, we compare video quality for SLA under zero-shot (0-step fine-tuning) and several smaller fine-tuning steps. We use a fixed top-k ratio of 5%, and keep all other settings the same as in the main paper. We observe that zero-shot replacement leads to clear quality degradation. However, after only 1K fine-tuning steps, the quality already becomes close to full attention, and 2K steps give the best results. We will add the experiment to the final manuscript.
>
> | Method          |  VA ↑ |  VT ↑ | IQ ↑ | OC ↑ | AQ ↑ | SC ↑ |    VR ↑ |
> | --------------- | ----: | ----: | ---: | ---: | ---: | ---: | ------: |
> | Full Attention  | 76.78 | 82.88 | 62.5 | 23.3 | 56.1 | 93.0 |   0.059 |
> | SLA (0 step)    | 41.11 | 51.79 | 58.3 | 21.4 | 46.7 | 81.0 | -0.1105 |
> | SLA (250 step)  | 64.46 | 78.06 | 59.0 | 22.8 | 55.7 | 88.5 | -0.0244 |
> | SLA (1000 step) | 74.58 | 80.09 | 61.8 | 23.7 | 56.1 | 92.3 |  0.0429 |
> | SLA (2000 step) | 76.96 | 83.92 | 62.2 | 23.6 | 55.9 | 93.1 |  0.0483 |
>
>
>
> ---
>
> > **W2.** Generalization beyond the tested architectures: Experiments focus on Wan2.1-1.3B and LightningDiT. Both employ similar attention layouts. It remains unclear whether SLA’s block classification and kernel fusion generalize to other DiT variants (e.g., different block sizes, mixed precision, cross-attention layers, text-video architectures). More diversity in model families (or a detailed discussion of expected behavior) would be valuable.
>
> **Reply:** Thank you for the suggestion. We are happy to provide more discussion and experiments:
>
> First, different block sizes and mixed precision are not DiT architecture variants. They only affect the implementation of attention. For cross-attention layers, the complexity is $O(N_t N_v d)$, which is much smaller than the self-attention cost $O(N_v^2 d)$ when $N_v \gg N_t$, where $N_t$ is the number of text tokens and $N_v$ is the number of video tokens. Thus, optimizing cross-attention is less important.
>
> Second, regarding DiT variants, the main families are standard DiT with cross attention and MM-DiT. Wan2.1 belongs to the DiT with cross attention architecture. To show that SLA also works on MM-DiT, we add an experiment on a private MM-DiT model. As shown in the Table below, using SLA with top-k = 5% and 2K fine-tuning steps, the MM-DiT model has almost no loss in end-to-end metrics compared to full attention.
>
> | Method         | VA ↑   | IQ ↑     | OC ↑     | AQ ↑     | SC ↑     |
> | -------------- | ------ | -------- | -------- | -------- | -------- |
> | Full Attention | 57.81 | 58.15 | 7.88 | 55.77 | 86.89 |
> | SLA            | 55.18 | 55.97 | 8.28 | 56.11 | 88.50 |
>
> This experiment confirms that SLA can be applied to MM-DiT, and we will add this experiment in our final paper.
>
>
>
> ---
>
> > **W3.** Ablation breadth: While Table 2 covers fusion, activation, and kh variations, the key hyper-parameter pair (kh, kl) and block size (bq, bkv) are not jointly explored. Similar attention heads with differing dimensions may require different thresholds; without a deeper ablation, reproducibility across setups could be challenging.
>
> **Reply:** Thank you for the insightful comment. To address this concern, we conduct additional experiments on the LightningDiT-1p0B/1 model to jointly examine the effects of the hyperparameter $k_l$ and block sizes $(b_q, b_{kv})$. The experimental settings are the same as those in our paper. The results are in the Table below. We can find that varying $b_q$ and $b_k$ leads to only marginal changes in performance, while increasing $k_l$ consistently degrades generation quality.
>
> | $b_q$ | $b_k$ | $k_l$    | FID ↓ |
> | ----- | ----- | -------- | ----- |
> | 64    | 64  | 0%    | 31.49 |
> | 128 | 64  | 0%    | 31.49 |
> | 128 | 32  | 0%    | 31.74 |
> | 64  | 64  | 6.25% | 32.94 |
> | 64 | 64 | 12.5% | 34.04 |
> | 64 | 64  | 25%   | 37.31 |

---

> ### Author Response · Authors · 2025-11-20
> **Authors’ Rebuttal-2 to the Weaknesses Raised by Reviewer PbE1**
>
> ---
>
> > **W4.** Evaluation scope: The paper reports strong aggregate metrics but lacks qualitative failure analysis. For instance, where does SLA still degrade outputs (e.g., motion consistency over long horizons, rare events)? Additionally, comparisons to other hybrid approaches (if any exist) or to recent structured sparsity methods in diffusion (e.g., block-diagonal or strided patterns) are absent.
>
> **Reply:** Thank you for the important question. We answer it step by step:
>
> 1. We collect two failure cases of SLA and put them in an anonymous link:
>    https://anonymous.4open.science/r/SLA_failure_case-F434
>    Both videos are in motion-consistency benchmark under very high attention sparsity (96%). From these cases, we find that SLA may fail when the sparsity is extremely high and the evaluation focuses on motion consistency. This might be related to the fact that linear attention has more difficulty in fully capturing complex video dynamics.
>
> 2. Thank you for the suggestion. We have compared SLA with the main trainable sparse attention methods (VSA, VMoBa, ...) on DiT model, and there is no existing hybrid method combining sparse and linear attention to the best of our knowledge. Additionally, as you suggest, we evaluate SLA against block-diagonal attention on the LightningDiT-1p0B/1 model. The results are in the Table below, showing that SLA consistently outperforms block-diagonal attention.
>
> | Method         | FID ↓ | FLOPs ↓ | Sparsity ↑ |
> | -------------- | ----- | ------- | ---------- |
> | block-diagonal | 55.06 | 3.22G   | 75.00%  |
> | SLA (ours)     | 31.49 | 1.73G   | 87%        |
>
>
>
> ---
>
> > **W5.** Kernel integration specifics: Although Appendix A.3 sketches optimization tricks, concrete profiling (e.g., breakdown of compute vs. memory bottlenecks, impact on backward pass peak memory) is missing. Readers aiming to reimplement SLA might struggle without more hardware-level detail.
>
> **Reply:** Thank you for the suggestion. We further analyze the kernel at top-k = 5% using the Nvidia ncu tool. The Table below reports the MFU, L2 cache I/O utilization, DRAM I/O utilization, and the peak memory usage (occupation) relative to FlashAttention for both the forward and backward passes. Both the theoretical analysis and the profiling results show that SLA is a compute-bound workload, without obvious memory bottlenecks. In addition, we plan to open-source our implementation after the paper is accepted, so readers can directly inspect and reuse our kernels instead of reimplementing them.
>
> | Kernel       | MFU   | L2 cache I/O Utilization | DRAM I/O Utilization | Memory Occupation |
> | ------------ | ----- | ------------------------ | -------------------- | ---------- |
> | SLA Foward   | 75.5% | 63.8%                    | 2.9%                 | 1.12x      |
> | SLA Backward | 54.2% | 57.4%                    | 1.8%                 | 1.15x      |
>
> ---
> **If you feel your concerns have been resolved, we would greatly appreciate it if you consider raising the score. We will add the experiment and analysis in the rebuttal to our paper.**

---

> ### Author Response · Authors · 2025-11-20
> **Authors’ Rebuttal to the Questions Raised by Reviewer PbE1**
>
> ---
> > **Q1.** Fine-tuning sensitivity: How does SLA perform if fine-tuning resources are restricted (fewer steps, smaller batch, or lower-resolution data)? Could you provide a curve illustrating quality vs. fine-tuning budget to justify the “few steps” claim quantitatively?
>
> **Reply:** Thank you for the suggestion. As discussed in our reply to W1, we add additional experiments to study SLA in the zero-shot and limited fine-tuning settings. As shown in the Table below, we compare video quality for SLA under zero-shot (0-step fine-tuning) and several smaller fine-tuning steps. We use a fixed top-k ratio of 5%, and keep all other settings the same as in the main paper. We observe that zero-shot replacement leads to clear quality degradation. However, after only 1K fine-tuning steps, the quality already becomes close to full attention, and 2K steps give the best results. We will add the experiment to the final manuscript.
>
> |Method|VA↑|VT↑|IQ↑|OC↑|AQ↑|SC↑|VR↑|
> |-|-|-|-|-|-|-|-|
> |Full Attention|76.78|82.88|62.5|23.3|56.1|93.0|0.059|
> |SLA (0 step)|41.11|51.79|58.3|21.4|46.7|81.0|-0.1105|
> |SLA (250 step)|64.46|78.06|59.0|22.8|55.7|88.5|-0.0244|
> |SLA (1000 step)|74.58|80.09|61.8|23.7|56.1|92.3|0.0429|
> |SLA (2000 step)|76.96|83.92|62.2|23.6|55.9|93.1|0.0483|
>
> ---
> > **Q2.** Mask prediction robustness: The compressed mask Mc relies on pooled Q/K projections. How sensitive is SLA to inaccuracies in $P_c$ during inference, especially when encountering distribution shifts (e.g., different video domains)? Have you evaluated adaptive thresholds or confidence measures?
>
> **Reply:** Thank you for the question. (1) In our experiments, SLA is quite robust to inaccuracies in $P_c$, and it behaves more stably than using a top-CDF rule. We compare the top-k and top-CDF strategies in the Table below. The results show that top-k consistently delivers better video quality, suggesting that SLA is not overly sensitive to small errors in $P_c$.
>
> (2) For distribution shifts across video domains, it is hard to give a precise quantitative conclusion now. So far, we have not observed a clear difference in the behavior of $P_c$ when moving between different video domains. We will further analyze and, if possible, include a more detailed study in the final paper.
>
> |Method|IQ↑|OC↑|AQ↑|SC↑|VR↑|
> |-|-|-|-|-|-|
> |SLA (top-k)|62.2|23.6|55.9|93.1|0.0483|
> |SLA (top-cdf)|60.8|22.2|53.8|91.2|-0.0263|
>
> ---
> > **Q3.** (1) Backward pass efficiency: The backward kernel reportedly yields a 6.8× speedup, but details are sparse. Can you share profiling numbers on gradient computation vs. memory traffic? (2) Are there cases where the backward pass becomes the new bottleneck? (3) Model generality: Have you attempted SLA on cross-attention or text-conditioned video generation models?
>
> **Reply:** Thank you for the suggestion. (1) We profile the backward pass. The Table below reports MFU, L2 cache I/O utilization, DRAM I/O utilization, and peak memory usage relative to FlashAttention. Both theory and profiling show that the SLA backward kernel is compute-bound, without clear memory bottlenecks, and the memory traffic is hidden by compute time.
>
> |Kernel|MFU|L2 cache I/O Utilization|DRAM I/O Utilization|Memory Use|
> |-|-|-|-|-|
> |SLA Backward|54.2%|57.4%|1.8%|1.15x|
>
> (2) We have not observed cases where the backward pass becomes the main bottleneck. We also want to emphasize that SLA is used mainly for inference acceleration. It only needs a few steps of fine-tuning, then it can be used for inference, so backward efficiency is less critical.
>
> (3) 1. We have not applied SLA to cross-attention layers yet, mainly because the time cost of cross-attention is small: cross-attention is typically much cheaper than self-attention in video generation. 2. For text-conditioned video generation, we have already evaluated SLA on a text-conditioned DiT model: Wan2.1 in the paper is exactly such a model.
>
> ---
> > **Q4.** Failure cases: Could you provide examples or statistics of prompts where SLA underperforms full attention (e.g., higher error metrics, visual artifacts)? Understanding the boundaries of applicability would help practitioners adopt the method safely.
>
> Reply: Thank you for the valuable suggestion. Please see the reply for W4.
>
> ---
> > **Q5.** Hyper-parameter tuning: For practitioners, how should kh and kl be selected for a new model? Do you recommend starting from a certain sparsity target and adjusting based on validation loss, or is there a principled heuristic?
>
> **Reply:** In practice, $k_l$ is a less important hyper-parameter and can be fixed to 5% or 0% by default. We can tune $k_h$, which mainly controls the overall sparsity and quality. A good starting point is $k_h$ = 10%; we recommend slightly increasing or decreasing $k_h$ to achieve the desired speed–quality trade-off.
>
> ---
> **If you feel your concerns have been resolved, we would greatly appreciate it if you consider raising the score. We will add the experiment and analysis in the rebuttal to our paper.**

---

### Official Review · Reviewer_fACC · 2025-10-31

**Soundness:** 3
**Presentation:** 3
**Contribution:** 3
**Rating:** 6
**Confidence:** 4

**Summary:**

This paper addresses the quadratic attention bottleneck in Diffusion Transformers (DiTs), particularly for video generation where sequence lengths are large. Based on the key observation that DiT attention weights can be decomposed into a small high-rank component (a few large weights) and a large, extremely low-rank component (many small weights), the authors propose Sparse-Linear Attention (SLA). SLA is a trainable, hybrid mechanism that classifies attention blocks into three categories: 1)Critical: Computed with $O(N^2)$ sparse attention. 2) Marginal: Computed with $O(N)$ linear attention. 3)Negligible: Skipped entirely.By fusing these operations into an efficient GPU kernel and applying minimal fine-tuning, SLA achieves a 95% reduction in $O(N^2)$ attention computation. This results in a 13.7x kernel speedup and a 2.2x end-to-end video generation speedup on the Wan2.1-1.3B model, all while maintaining generation quality.

**Strengths:**

1. The core observation decoupling attention weights into a sparse/high-rank component and a dense/low-rank component is novel and insightful. It clearly explains the limitations of using only sparse or only linear attention and provides a strong foundation for the proposed hybrid approach
2. The proposed SLA method, which dynamically assigns different computational complexities ($O(N^2)$, $O(N)$, or $O(0)$) based on a block's predicted importance, is an elegant and highly effective solution.
3. The paper demonstrates truly significant gains. A 95% reduction in expensive $O(N^2)$ computation translating to a 13.7x kernel speedup and a 2.2x end-to-end speedup with no loss in quality is a major contribution for efficient DiTs

**Weaknesses:**

1. The "95% sparsity" figure is confusing. Based on the hyperparameters ($k_h$=5%, $k_l$=10%), this 95% refers to blocks that avoid $O(N^2)$ computation. It does not mean 95% of blocks are skipped. In fact, 85% (100% - 5% - 10%) are still processed by $O(N)$ linear attention. This terminology should be clarified
2. All latency benchmarks (e.g., 13.7x speedup) are reported on an RTX 5090 GPU. While impressive, providing benchmarks on more widely accessible hardware  would greatly strengthen the paper's practical and immediate value to the research community.

**Questions:**

See Weaknesses

---

> ### Author Response · Authors · 2025-11-20
> **Authors’ Rebuttal to the Weaknesses Raised by Reviewer fACC**
>
> Dear Reviewer fACC,
> Thank you for your valuable suggestions and questions. Below, we address each point raised.
>
> ---
>
> > **W1.** The "95% sparsity" figure is confusing. Based on the hyperparameters ($k_h$=5%, $k_l$=10%), this 95% refers to blocks that avoid $O(N^2)$ computation. It does not mean 95% of blocks are skipped. In fact, 85% (100% - 5% - 10%) are still processed by $O(N)$ linear attention. This terminology should be clarified.
>
> **Reply:** Thank you so much for the suggestion. We agree that the term “95% sparsity” can be confusing, and we will revise the final manuscript to clarify that it refers to the proportion of blocks that avoid $O(N^2)$ attention, and state how many blocks use linear attention and what their approximate complexity is. In our setting, the cost of linear attention is much smaller than that of sparse or full attention. For example, when N = 32K, d = 128, we have $O(N d^2) \ll O(N^2 d)$, the extra cost from the linear part is negligible in practice ($O(N d^2)$ =0.004 $\times$ $O(N^2 d)$). In this case, 95\% sparsity in the sparse attention component means 94.7\% attention complexity reduction. **We will revise our paper to clarify this point, as you suggested.**
>
>
>
> ---
>
> > **W2.** All latency benchmarks (e.g., 13.7x speedup) are reported on an RTX 5090 GPU. While impressive, providing benchmarks on more widely accessible hardware would greatly strengthen the paper's practical and immediate value to the research community.
>
> **Reply:** Thank you for the insightful suggestion. We conduct additional experiments on more GPUs, including the RTX4090, A800, L20, and H800. The efficiency of SLA under 95% sparsity is shown in the Table below:
>
> | GPU     | Speedup |
> | ------- | ------- |
> | RTX4090 | 11.20x  |
> | A800    | 10.76x   |
> | L20     | 13.18x  |
> | H800    | 12.94x  |
>
> ---
>
> **If you feel your concerns have been resolved, we would greatly appreciate it if you consider raising the score. We will add the experiment and analysis in the rebuttal to our paper.**

---

### Official Review · Reviewer_kaNn · 2025-10-31

**Soundness:** 2
**Presentation:** 2
**Contribution:** 2
**Rating:** 2
**Confidence:** 5

**Summary:**

This work adopts the linear attention for the intermediate stage between the sparse attention and full (dense) attention.

This work first adopts the pooling to $Q$ and $K$ to get the sparse mask, then apply the full attention to the important area, skip the unimportant area, and adopt linear attention to those intermediate important area.

This work adopt the fine-tuning for the new introduced linear attention weights, which I think is the main contribution of this work. This work makes a lot effort on the forward and backward implementation of the linear attention for the insert to the original full attention and sparse attention technique.

according to the results, this work achieves good results.

**Strengths:**

1. This work makes a lot of effort for the implementation of the linear attention - the forward and backward - during the fine-tuning.

**Weaknesses:**

1. The equation (2) which adopts the mean pooling to identify the attention importance is the same as the work [1] which also use the mean pooling to identify the attention importance for the acceleration of video generation. Thus, I think the novelty is limited to this work.

2. This work does not provide any similarity metric like PSNR, SSIM, and LPIPS in their results, then the comparison to the work [1] is not direct. As the work [1] also use the sparse attention for the acceleration, and achieves 90% sparsity, so the exact performance improvement from the linear attention remains unknown.

3. This work focus more on the new CUDA kernel designs rather than some algorithm-level novelty.

---

[1] DraftAttention: Fast Video Diffusion via Low-Resolution Attention Guidance

**Questions:**

1. This work may focus on the contribution of the implementation of the forward and backward for the insert of linear attention to the full attention and sparse attention, then I think the novelty is limited.

2. Can the authors provide the similarity metric like PSNR, SSIM, and LPIPS to compare to other works like [1] [2]. Especially, compare to the work [1]. Theoretically, the linear attention is slower than the skip (i.e., the sparse attention) and this work takes a lot of training effort for fine-tuning, authors need to show that the linear attention exactly improve much performance than the pure sparse attention.

3. According to the result in work [1] (Table 1), the 90% sparsity ratio does not lead to much accuracy drop on image quality, while the result in Table 2 of this work show that the 'Sparse Only' result shows a large accuracy drop on metric like 'Image Quality'. Can the author explain this?

4. For this work, which adopts the linear attention for the intermediate stage between full and sparse attention, I would like to ask for the results in low-resolution. I think it may be enough to adopt the sparse attention for acceleration in low-resolution video generation, as there may not be so many 'Marginal' blocks in the low-resolution video generation process.

5. I want to further ask the authors about: the fine-tuning results for the pure sparse attention technique with downsampled attention (i.e., equation (2) in this paper). I think it is quite important ablation to show the effectiveness of the linear attention under the same training effort. I believe if apply same fine-tuning effort to pure sparse attention, the results will be good.

---
[1] DraftAttention: Fast Video Diffusion via Low-Resolution Attention Guidance

[2] Sparse VideoGen: Accelerating Video Diffusion Transformers with Spatial-Temporal Sparsity

---

> ### Author Response · Authors · 2025-11-20
> **Authors’ Rebuttal to the Weaknesses Raised by Reviewer kaNn**
>
> Dear Reviewer kaNn,
> Thank you for your valuable suggestions and questions. Below, we address each point raised.
>
> ---
>
> >  **W1.** The equation (2) which adopts the mean pooling to identify the attention importance is the same as the work [1] which also use the mean pooling to identify the attention importance for the acceleration of video generation. Thus, I think the novelty is limited to this work.
>
> **Reply:** **We do not claim the mean pooling as our contribution.** Actually, it is only a small detail in our method. Our contributions are:
>
> - We find that the attention maps in diffusion models can be perfectly decomposed into two parts: 1) a very sparse but high-rank part, and 2) a very low-rank but dense part.
> - We propose the first attention method that effectively fuses sparse attention and linear attention.
> - Our method achieves 95% attention sparsity, corresponding to approximately a 20× reduction in attention computation, while maintaining video generation quality.
> - We implement efficient GPU kernels for SLA.
>
>
>
> ---
>
> > **W2.** (1) This work does not provide any similarity metric like PSNR, SSIM, and LPIPS in their results, then the comparison to the work [1] is not direct. (2-1) As the work [1] also use the sparse attention for the acceleration, and achieves 90% sparsity, (2-2) so the exact performance improvement from the linear attention remains unknown.
>
> **Reply:** Thank you for your questions. We address them one by one:
>
> (1) PSNR, SSIM, and LPIPS are mainly similarity metrics for video: they are **only suitable in training-free settings**, where the model parameters are fixed and only measure how close the generated videos are to those from originals. However, in our method, we need to fine-tune the model parameters, so the generated content will naturally change. In this case, PSNR, SSIM, and LPIPS are no longer suitable as the main metrics. Instead, it is more appropriate to use video quality metrics.
>
> (2-1) The 90% sparsity in [1]-Table1 is reported on Hunyuan Video at 720p resolution, where the attention sequence length is about 120K tokens, much larger than the 32K tokens for Wan2.1-1.3B in our setting. **Importantly, the results in [1]-Table1 also confirm this: on lower-resolution Wan2.1 videos, [1]'s sparsity only reaches up to 75%, while on the same Wan2.1 model, SLA achieves a sparsity of 95%.**
>
> (2-2) In **Table 2 of our original paper**, we provide a detailed ablation between Sparse Attention Only and SLA. The results clearly show that the “Sparse Only” variant performs worse, while SLA gives the best performance. This **demonstrates that the improvement from adding linear attention is significant.**
>
>
>
> ---
>
> > **W3.** This work focus more on the new CUDA kernel designs rather than some algorithm-level novelty.
>
> **Reply:** **Our main contributions are at the algorithm level.** The kernel implementation is a minor part of our contribution. Specifically, our contributions are:
>
> - We find that the attention maps in diffusion models can be perfectly decomposed into two parts: 1) a very sparse but high-rank part, and 2) a very low-rank but dense part.
> - We propose the first attention method that effectively fuses sparse attention and linear attention.
> - Our method achieves 95% attention sparsity, corresponding to approximately a 20× reduction in attention computation, while maintaining video generation quality.
> - We implement efficient GPU kernels for SLA.

---

> ### Author Response · Authors · 2025-11-20
> **Authors’ Rebuttal to the Questions Raised by Reviewer kaNn**
>
> ---
> > **Q1.** This work may focus on the contribution of the implementation of the forward and backward for the insert of linear attention to the full attention and sparse attention, then I think the novelty is limited.
>
> **Reply:** **Our main contributions are at the algorithm level.** The kernel implementation is a minor part of our contribution. Specifically, our contributions are:
>
> - We find that the attention maps in diffusion models can be perfectly decomposed into two parts: 1) a very sparse but high-rank part, and 2) a very low-rank but dense part.
> - We propose the first attention method that effectively fuses sparse attention and linear attention.
> - Our method achieves 95% attention sparsity, corresponding to approximately a 20× reduction in attention computation, while maintaining video generation quality.
> - We implement efficient GPU kernels for SLA.
>
> ---
> > **Q2.** (1) Can the authors provide the similarity metric like PSNR, SSIM, and LPIPS to compare to other works like [1] [2]. (2) Especially, compare to the work [1]. Theoretically, the linear attention is slower than the skip (i.e., the sparse attention) and this work takes a lot of training effort for fine-tuning, authors need to show that the linear attention exactly improve much performance than the pure sparse attention.
>
> **Reply:** (1) PSNR, SSIM, and LPIPS are mainly similarity metrics for video: they are **only suitable in training-free settings**, where the model parameters are fixed and only measure how close the generated videos are to those from originals. However, in our method, we need to fine-tune the model parameters, so the generated content will naturally change. In this case, PSNR, SSIM, and LPIPS are no longer suitable as the main metrics. Instead, it is more appropriate to use video quality metrics.
>
> (2) First, the computation cost of linear attention is much lower than that of sparse or full attention. For example, in our setting with sequence length N > 32K, d=128, the complexity of the linear part $O(N d^2)$ is much smaller than the original attention cost $O(N^2 d)$, ($O(N d^2)$ =0.004 $\times$ $O(N^2 d)$), so the extra cost of linear attention is negligible. Second, **Table 2 in our original paper** gives a detailed ablation between the “Sparse Attention Only” variant and SLA. The “Sparse Only” model has clearly worse performance, while SLA performs best. This shows that **the improvement from adding linear attention is significant.**
>
> ---
> > **Q3.** According to the result in work [1] (Table 1), the 90% sparsity ratio does not lead to much accuracy drop on image quality, while the result in Table 2 of this work show that the 'Sparse Only' result shows a large accuracy drop on metric like 'Image Quality'. Can the author explain this?
>
> **Reply:** In [1]-Table1, the 90% sparsity is obtained on Hunyuan Video at 720p resolution, where the attention sequence length is about 120K tokens. This is much larger than the 32K sequence length of Wan2.1-1.3B at 480p used in our experiments. It is easier to reach higher sparsity when the sequence is longer. **Importantly, the results in [1]-Table1 also confirm this: on lower-resolution Wan2.1 videos, [1]'s sparsity only reaches up to 75%, while on the same Wan2.1 model, SLA achieves a sparsity of 95%.**
>
> ---
> > **Q4.** For this work, which adopts the linear attention for the intermediate stage between full and sparse attention, I would like to ask for the results in low-resolution. I think it may be enough to adopt the sparse attention for acceleration in low-resolution video generation, as there may not be so many 'Marginal' blocks in the low-resolution video generation process.
>
> **Reply:** First, our experiments are conducted on **480p videos, which is already the lowest resolution available** in our setting. Second, as shown in **Table 2 of our original paper**, using **only high-sparsity sparse attention at this resolution is insufficient, and the additional linear attention clearly helps** maintain video quality in a high attention sparsity.
>
> ---
> > **Q5.** I want to further ask the authors about: the fine-tuning results for the pure sparse attention technique with downsampled attention (i.e., equation (2) in this paper). I think it is quite important ablation to show the effectiveness of the linear attention under the same training effort. I believe if apply same fine-tuning effort to pure sparse attention, the results will be good.
>
> **Reply:** We already provide this ablation in **Table 2 of our original paper**. The “Sparse Only” variant is exactly the pure sparse attention using Eq. (2), trained under the same fine-tuning setup as SLA (same number of steps, dataset, learning rate, etc.). The results show that the “Sparse Only” performs worse, **confirming the benefit of adding linear attention.**
>
> ---
> **If you feel your concerns have been resolved, we would greatly appreciate it if you consider raising the score. We will add the analysis in the rebuttal to our paper.**

---

> > ### Comment · Reviewer_kaNn · 2025-11-24
> >
> > 1. For W2 and Q2:
> >
> > The reason why I asked the question about the similarity metrics is authors show the visualization results in Figure 5. That is, authors show that the 95% SLA results can achieve similar results with the Full Attention, where I think the similarity results should work for the dense inference or sparse inference with the fine-tuned model. Meanwhile, as the authors are claiming that the low-rank but dense part (linear attention part) is decomposed from the original attention map, then I think the sparse inference should generate the similar results as the dense inference as shown in Figure 5.
> >
> > The above is my motivation of asking for the similarity metrics including PSNR, SSIM, and LPIPS. If the results are similar between dense inference and sparse inference as shown in Figure 5, the requirement of similarity metrics is needed for the verification of the effectiveness in this paper, especially the comparison to other works like [1] [2]. Besides, according to Table 1 in work [1], the results of similarity metrics show much different accuracy drop from the video quality. For example, for high sparsity 90% ratio of Hunyuan model results in Table 1 of work [1], the similarity metrics including PSNR, SSIM, and LPIPS drops a lot when the Image Quality remains stable.
> >
> > Therefore, This is **main concern** of this paper that does not provide the similarity metrics.
> >
> >
> > 2. For Q3:
> >
> > Can the authors explain the exact sparsity ratio and the linear attention computation cost used for results? It may not be fair to compare to work [1] which adopts the whole sparse attention with 75% sparsity ratio, when authors include the fine-tuned linear attention.
> >
> >
> > 3. For Q4 & Q5:
> >
> > According to the author rebuttal, the 'Sparse Only' results in Table 2 achieves **with the fine-tuning** when pure sparse attention is applied. Then, the results of 'Image Quality' is 57.2 (when the dense is 62.5), which is too low compared to the results in Table 1 of work [1] (i.e., the fine-tuning may not improve the generation performance). I think such results may come from that, the fine-tuning strategy for pure sparse attention method may not be suitable, as it is used the same way as the mix (sparse attention and linear attention) strategy. Thus, such results may not be enough to show the effective ablation of the proposed method.
> >
> > ---
> >
> > Meanwhile, another question is: does this method still effective for high-resolution video generation? I mean the generalization of the method.
> >
> > I know this question is wired, that is because I just understood the results were achieved for high-resolution results and might not be generalized to the low resolution tasks. However, according to the author rebuttal, they said in (2) of rebuttal for Q2, the linear attention costs much less computation cost when the number of token sequence becomes large, and for high-resolution tasks, the number of token sequence would be more than 100k, I think the linear attention may fail.
> >
> > ----
> >
> > [1] DraftAttention: Fast Video Diffusion via Low-Resolution Attention Guidance
> >
> > [2] Sparse VideoGen: Accelerating Video Diffusion Transformers with Spatial-Temporal Sparsity

---

> ### Author Response · Authors · 2025-11-24
> **Authors’ Rebuttal to the further questions of Reviewer kaNn**
>
> We thank reviewer kaNn for the additional questions. We greatly appreciate the opportunity to further clarify, and we address each additional question below.
>
> ---
>
> > **1.** Reply to the further questions on W2 and Q2:
>
> Thank you for the suggestion on using PSNR, SSIM, and LPIPS. We still argue that these similarity metrics are only appropriate for training-free settings: (but we still provide the similarity results in final)
>
> - First, in our setting, the model is fine-tuned, and its outputs are not expected to be pixel-wise similar to those of the full-attention model. For example, in Figure 5, the second pair is only similar in semantics but different in content.
>
> - Second, in the anonymous link (https://anonymous.4open.science/r/SLA_rebuttal_videos-4DF4), we provide more examples comparing SLA and full attention. In many cases, the videos are semantically consistent and of high quality, but the exact visual content differs substantially.
>
> Therefore, we believe that forcing high PSNR/SSIM/LPIPS with respect to full-attention outputs is not necessarily aligned with better video generation quality, once the model has been fine-tuned. The goal of fine-tuning is to preserve semantic quality under sparsity, not to reproduce the exact dense-attention frames.
>
> ---
> Finally, to provide the requested experiment results, we compute PSNR, SSIM, and LPIPS between SLA and full attention. The results are shown in the Table below:
>
> | PSNR    | SSIM    | LPIPS    |
> | ------- | ------- | -------- |
> | 17.3625 | 0.67436 | 0.298275 |
>
> ---
>
> > **2.** Response to the further question on Q3:
>
> Thank you for the question. The exact sparsity ratio and the linear attention computation cost used for the results is:
>
> * The sparse attention part uses a sparsity ratio of 95%, i.e., it retains 5% of the full-attention computation.
> * The linear attention part contributes an additional 0.4% of the full-attention computation cost.
>
> We would also like to emphasize that we do not claim that our comparison to [1] is strictly “fair”. The comparison was proposed according to the reviewer’s request, rather than as a central claim of our paper.
>
>
>
> ---
>
> > **3.** Response to the further questions on Q4 and Q5:
>
> Thank you for the further questions. However, we believe it is **not justified** to conclude that “fine-tuning may not improve the generation performance” based solely on comparing our sparse-only results to those reported in [1]. There are two reasons:
>
> * The “Sparse Only” configuration in our Table 2 uses 85% sparsity. There is no directly comparable result in Table 1 of [1] under the same model and the same sparsity level.
> * Moreover and importantly, the "sparse-only" method in our paper is not identical to the sparse attention method in [1]. The fact that [1] reports little quality drop may be due to differences in the specific method, and therefore, the two sets of results cannot be directly compared.
>
> Furthermore, Table2 does provide a meaningful ablation: under the same model, data, and fine-tuning settings, we consistently observe: Sparse + Linear > Sparse Only > Linear Only, which supports the effectiveness of combining sparse attention with linear attention.
>
> ---
>
> > **4.** Response to the another question on high-resolution video generation:
>
> Yes, SLA is still effective for high-resolution video generation.
>
> We add an experiment on an MM-DiT model for 720p video generation. As shown in the table below, with SLA, top-k = 5% and 2K fine-tuning steps, the MM-DiT model shows effectiveness.
>
> | Method         | VA ↑      | SC ↑    | IQ ↑     | OC ↑     | AQ ↑     |
> | -------------- | -------- | -------- | -------- | -------- | -------- |
> | Full Attention | 0.868938  | 57.814  |  0.581483 | 0.078805 | 0.557694 |
> | SLA           | **0.884979**  | 55.177   |  0.559733 | **0.082825** | **0.561116** |
>
> ---
> Thank you again for your time, and we sincerely look forward to your reply.

---

> > ### Comment · Reviewer_kaNn · 2025-11-25
> >
> > 1. I am not satisfied with the similarity results. The full attention results are generated by the original base model or the fine-tuned model with dense inference?
> >
> > 3. If the authors claim the *the "sparse-only" method in our paper is not identical to the sparse attention method in [1].*, then what is the sparse attention method authors used for 'sparse-only'? According to the Section 4 and Section 4.1 with Eq (2) (3) (4), I believe authors are using the same sparse attention method with work [1]. Besides, authors adopt the fine-tuning for the model and still achieved worse results, which is not acceptable.
> >
> > 4. For the high-resolution task, if the sparsity is 95%? The Image Quality metric drops a lot compared to the low-resolution task.
> >
> >
> > [1] DraftAttention: Fast Video Diffusion via Low-Resolution Attention Guidance

---

> ### Author Response · Authors · 2025-11-26
> **Authors’ Rebuttal to the further questions of Reviewer kaNn**
>
> We address each point raised.
>
> ---
> 1. We have already explained why the similarity metric is only appropriate for training-free settings rather than the fine-tuning settings we use. The video comparison examples in the prior reply (https://anonymous.4open.science/r/SLA_rebuttal_videos-4DF4
> ), together with the similarity results, **both exactly support this claim.**
>
> ---
> 2. Our “sparse-only” setting is, **of course, not identical** to the sparse attention method in [1]. For example, [1] also applies token reordering on the query/key/value matrices, which is different from the "sparse-only" baseline.
>
> ---
> 3. Yes, the sparsity is 95%. Importantly, if we check the average degradation over all evaluation metrics, SLA only introduces about a 0.05% loss in end-to-end quality, which is very small. This still demonstrates that SLA remains effective for high-resolution generation. Moreover, we will further refine this experiment in the final version, as the rebuttal period is limited.

---

> > ### Comment · Reviewer_kaNn · 2025-11-26
> >
> > Thank you for your rebuttal.
> >
> > 1. My concern is from the Figure 5, authors show the identical results in pixel level, which motivates me to ask for the results of the similarity metrics. However, authors' rebuttal shows that there is no pixel level similarity, which is a conflict. Meanwhile, I am still inquisitive about the previous question: *The full attention results are generated by the original base model or the fine-tuned model with dense inference?*
> >
> > 2. After reviewing the paper [1], I believe the *token reordering* is designed and adopted for the hardware efficiency, which is **not related to the generation performance**. Thus, I disagree with the authors on this point. Besides, I want to ask that, the work [1] adopts the token reordering in their work to group the dense part and sparse part, separately, for faster generation, then, how does this work SLA to solve this problem? If SLA does not adopt similar technique as token reordering, then the block size of the sparse attention may be much smaller than the work [1]. Can the authors provide results for the block size used for sparse attention? and also some ablation study?
> >
> > 3. For video quality metrics, it is not sensitivity to the generation performance, which can be verified by the works [1] [2]. In Table 1 of both works [1] [2], the video quality metrics only drop a little when the similarity metrics drops a lot. Therefore, considering the Image Quality drops a lot for high-resolution task, I disagree with the authors' point that the SLA is still effective in high-resolution task.
> >
> > -----
> >
> > [1] DraftAttention: Fast Video Diffusion via Low-Resolution Attention Guidance
> >
> > [2] Sparse VideoGen: Accelerating Video Diffusion Transformers with Spatial-Temporal Sparsity

---

> ### Author Response · Authors · 2025-11-29
>
> ---
> 1. Only one video in Figure 5 shows identical results at the pixel level. The **other** video in Figure 5 does not show this property. In addition, we provide **more comparison examples at** (https://anonymous.4open.science/r/SLA_rebuttal_videos-4DF4), showing that the videos are only semantically consistent, without pixel-wise identical results.
>
> ---
> 2. Token reordering is **not only a hardware implementation**. From an algorithmic perspective, token reordering can **achieve higher attention sparsity** under the same attention accuracy.
>
> ---
> 3. We further provide the image quality of 720p video generation under SLA (top-k = 5% and 3K fine-tuning steps) as shown in the following table. It can be seen that SLA does not cause a significant loss in IQ.
>
>
> | Method | IQ ↑ |
> | --- | --- |
> | Full Attention | 0.581483 |
> | SLA | 0.578739 |
>
> ---
> **Finally and most importantly, we find that the official review policy of ICLR 2026 (https://iclr.cc/Conferences/2026/ReviewerGuide) explicitly states that one should not compare against papers that are solely on arXiv, i.e., not published. We find that [1] is solely on arXiv and not published. Therefore, all comparisons with [1] violate the review policy.**
>
> ---
> [1] DraftAttention: Fast Video Diffusion via Low-Resolution Attention Guidance

---

### Author Response · Authors · 2025-11-20
**Rebuttal by Authors: Main Contributions and Responses to Common Questions**

Dear reviewers, thank you very much for your time and helpful comments.
Before addressing each reviewer's specific concerns, we would ike to highlight our main contributions and then answer several common questions.

---

> **Main contributions:**

- We find an important observation: attention maps in diffusion models can be perfectly decomposed into two parts: 1) a very sparse but high-rank part, and 2) a very low-rank but dense part.
- We propose the first attention method that effectively fuses sparse attention and linear attention.
- Our method achieves 95% attention sparsity, corresponding to approximately a 20× reduction in attention computation, while maintaining video generation quality.
- We implement efficient GPU kernels for SLA.

---

> **Question1. Is 95% sparsity really significant?**

**Reply:** Yes. The 95% sparsity is reported on Wan2.1-1.3B, at 480p resolution and 5-second videos, where the sequence length is about **32K** tokens. For this sequence length, previous work on sparse attention for video diffusion achieves sparsity levels of approximately 75% at most. Reaching 95% sparsity at this sequence length, while keeping quality, is a significant improvement because it reduces the remaining attention computation from 25% to 5%, a **5× improvement**.

---

> **Question2. Does 95% sparsity mean about 95% computation saving? How about the cost of linear attention?**

**Reply:** In our setting, 95% sparsity is roughly equal to a 95% reduction in attention computation. As analyzed in our original paper (around Line 230), the total cost of SLA can be written as: (1) Sparse attention: complexity $O(N^2 d) \times (1-\text{sparsity})$. (2) Linear attention: complexity $O(N d^2)$.

When the sequence length $N$ is large (e.g., $N = 32\text{K}$), we have $O(N d^2) \ll O(N^2 d)$, so the cost of the linear attention part is much smaller than the original full attention cost. With a more precise calculation, about 95% sparsity corresponds to a 94.7% reduction in attention computation. We will revise our paper to clarify this point.

---
Below, we provide point-by-point responses to each reviewer’s comments. We hope these address your concerns and would be happy to provide further clarification if needed.

---

### Meta-Review · Area_Chair_owjs · 2026-01-07

**Summary:**

Reviewers appreciated that the proposed method provides an effective solution. Reviewers also appreciated the originality of the approach, the engineering effort, and found to the paper to be well-written. Reviewers also raised some concerns like comparison with DraftAttention (Arxiv 2025), benchmarking on more widely accessible hardware, generalization beyond the tested architectures, missing ablations (e.g., hyper-parameter pair (kh, kl) and block size (bq, bkv) are not jointly explored), and additional implementation details and evaluation results.

**Reviewer Concerns:**

The meta-reviewer believes that the authors provided a comprehensive rebuttal. Despite DraftAttention (Arxiv 2025) being an unpublished arxiv work, authors provided a detailed comparison of their proposed approach with DraftAttention to address reviewer kaNn main concern. Authors provided justification on using similarity metrics, additional results based on PSNR, SSIM, and LPIPS to compare SLA and full attention, and clarification regarding both technical contributions and implementation efforts with respect to efficient GPU kernels for SLA. In the rebuttal, authors also provided details on the motivation of Sparse-Linear Attention, a fine-tuning sensitivity analysis, additional ablations requested by reviewers, 720p video generation results on an MM-DiT model, and hardware generality experiments. Post-rebuttal, there has been discussion between the authors and reviewer kaNn which indicates that the concerns related to comparison with  DraftAttention were not fully addressed as per this reviewer.

**Reviewer Scores:**

The meta-reviewer believes most of the concerns of the three reviewers (fACC, PbE1, and 24AN) were addressed in the rebuttal. These reviewers were already on the positive side. However, the meta-reviewer believes the fourth reviewer would have still remained negative based on the post-rebuttal discussion. Given that most of the reviewers are on the positive side, a comprehensive rebuttal provided by the authors, and post-rebuttal discussion regarding differences to DraftAttention (despite DraftAttention being an unpublished work), the meta-reviewer believes the strength outweights weaknesses for this manuscript and agrees with the majority of the reviewers positive ratings.

---

### Decision · Program_Chairs · 2026-01-26

Accept (Poster)